# Light-enhanced molecular polarity enabling multispectral color-cognitive memristor for neuromorphic visual system

Jongmin Lee[1,2], Bum Ho Jeong[1,2], Eswaran Kamaraj [3], Dohyung Kim[1,2], Hakjun Kim[1,2], Sanghyuk Park [3] ✉ & Hui Joon Park [1,2,4] ✉

An optoelectronic synapse having a multispectral color-discriminating ability is an essential prerequisite to emulate the human retina for realizing a neuromorphic visual system. Several studies based on the three-terminal transistor architecture have shown its feasibility; however, its implementation with a two-terminal memristor architecture, advantageous to achieving high integration density as a simple crossbar array for an ultra-high-resolution vision chip, remains a challenge. Furthermore, regardless of the architecture, it requires specific material combinations to exhibit the photo-synaptic functionalities, and thus its integration into various systems is limited. Here, we suggest an approach that can universally introduce a color-discriminating synaptic functionality into a two-terminal memristor irrespective of the kinds of switching medium. This is possible by simply introducing the molecular interlayer with long-lasting photo-enhanced dipoles that can adjust the resistance of the memristor at the light-irradiation. We also propose the molecular design principle that can afford this feature. The optoelectronic synapse array having a color-discriminating functionality is confirmed to improve the inference accuracy of the convolutional neural network for the colorful image recognition tasks through a visual pre-processing. Additionally, the wavelength-dependent optoelectronic synapse can also be leveraged in the design of a light-programmable reservoir computing system.

The human visual system is an essential element of human body that enables us to perceive visual information from the complicated outside world[1]. Especially, the retina in the human eyes has been known to perform crucial roles in the visual perception by detecting and first-stage pre-processing the light signals before complex high-level signal processing in the visual cortex of the brain[2,3]. In this context, the retina not only converts the visual input into the electrical signals, which travel to the brain, using the photoreceptor, but also extracts main characteristics from massive input signals (e.g., filtering, amplification, adaptation, and memory) by pre-processing those signals through the synapses connecting the neurons—photoreceptor, horizontal, bipolar, amacrine, and ganglion cell, which form a hierarchical biostructure[4]. This pre-process reduces redundant visual input data and accelerates complex data processing in the brain, such as image cognition, learning, and interpretation[5]. Therefore, to successfully emulate the retina and thus realize the efficient artificial visual system, the optoelectronic devices that can convert the light signals into the electrical signals, even with synaptic plasticity contributing to local computation of visual information as pre-image-processing, should be demonstrated.

[1]Department of Organic and Nano Engineering, Hanyang University, Seoul 04763, Republic of Korea. [2]Human-Tech Convergence Program, Hanyang University, Seoul 04763, Republic of Korea. [3]Department of Chemistry, Kongju National University, Kongju 32588, Republic of Korea. [4]Hanyang Institute of Smart Semiconductor, Seoul 04763, Republic of Korea. ✉e-mail: spark0920@kongju.ac.kr; huijoon@hanyang.ac.kr

One of the crucial functionalities of the retina enabling these roles is its ability to recognize the colorful information (e.g., by cone photoreceptors). Thus, the optoelectronic synapse devices are further required to possess a multispectral color-discriminating ability for achieving the artificial visual system. This feature is additionally beneficial to reducing the size and complexity of each pixel, and in this perspective a simple two-terminal architecture (e.g., memristor) having a merit in achieving high integration density as a crossbar array is much more favorable to achieving an ultra-high-resolution retinomorphic vision chip than a three-terminal transistor architecture facing the limitations on the device-downscaling[6–8].

Given these factors, various strategies that can overcome the limitation of the early-stage research works, which simply integrate a photodetector with a synaptic device[9,10], complicating the circuitry, have been proposed (Supplementary Table 1); however, approaches exhibiting the color-discriminating ability were mainly restricted to the three-terminal architecture[11–14], and the realization of color-distinguishable mechanism with a two-terminal memristor, competitive to enhancing its integration density even with its excellent electrical characteristics such as low operating voltage and high switching speed[15–17], still remains a challenge. Furthermore, regardless of the architectures, most of works in literature inevitably require specific material combinations as their mediums (e.g., for absorbing photon[13,18–21], tuning defect density[18,20], inducing charge-trapping[19,21,22], injecting carrier[23], controlling doping[24], varying valence state[25], etc.) to exhibit their photo-synaptic functionalities, and thus integration of those devices into various systems is hampered.

In this work, we suggest an approach that can introduce a color-discriminating synaptic functionality into a simple two-terminal memristor. Particularly, this mechanism is not limited to the memristors, where specific material combinations are essential for their functionalities. By inserting an organic semiconductor thin film, composed of asymmetric molecule having high dipole moment in the photo-excited state, between an electrode and a switching medium, the electric field within the device is confirmed to be enhanced at the light-irradiation, which means that light can be utilized as an input signal for adjusting the conductance of the memristor instead of the voltage bias. This photo-functionality of the molecules can be maximized by designing them to retain their enhanced dipole moments at the exited state with an extended lifetime, and its dependence on the spectral absorbance of the molecules help a single pixel to have the color selectivity even without any optical filters and complicated circuitry. The optoelectronic synapse array having this color-discriminating functionality is confirmed to improve the inference accuracy of the convolutional neural network (CNN) about the colorful image recognition tasks through a visual pre-processing. In addition, the wavelength-dependent optoelectronic synapse can also be leveraged in the design of a light-programmable reservoir computing system.

## Results

### Design of organic molecular structure

Figure 1a–h are molecular structures of the organic semiconductors to propose the mechanism of the optoelectronic memristor synapse with a multispectral color-recognition capability in a single pixel, which are 4-(di(naphthalen-2-yl)amino)-2-(1,4,5-triphenyl-1H-imidazol-2-yl)phenol, N-(naphthalen-2-yl)-N-(3-(1,4,5-triphenyl-1H-imidazol-2-yl)phenyl)naphthalen-2-amine, 2-(4,5-bis(4-fluorophenyl)-1-phenyl-1H-imidazol-2-yl)-4-(di(naphthalen-2-yl)amino)phenol, N-(3-(4,5-bis(4-fluorophenyl)-1-phenyl-1H-imidazol-2-yl)phenyl)-N-(naphthalen-2-yl)naphthalen-2-amine, (E)-3-(4-(dimethylamino) phenyl)-1-(1-hydroxynaphthalen-2-yl)prop-2-en-1-one, (E)-3-(4-(dimethylamino) phenyl)-1-(naphthalen-2-yl)prop-2-en-1-one, (E)-3-(4-(diphenylamino)phenyl)-1-(1-hydroxynaphthalen-2-yl)prop-2-en-1-one, and (E)-3-(4-(diphenylamino)phenyl)-1-(naphthalen-2-yl)prop-2-en-1-one,

denoted as DNH, DN, DNH-F, DN-F, CH-M, C-M, CH-P, and C-P. As a proof of concept, a synaptic memristor having photo-responsivity in UV wavelength region is demonstrated first, and then this architecture is further extended to realize the optoelectronic synapse with a color distinction ability in visible region. For this purpose, the first step is to control the bandgap of the material to respond to a specific wavelength. Next, the materials should exhibit a high intrinsic dipole moment upon excitation, and the excited state lifetime should be extended to ensure the effectiveness of the excited state. In this section, we will focus on the design principles to control the bandgap with enhanced intrinsic dipole moment, while the design for extended lifetime at the excitation will be discussed in detail in the next section, along with the operational principle of the memristor.

Three different material design strategies—an enhancement of the donor–acceptor (D-A) strength, an extension of the effective conjugation length, and an introduction of nodal plane model[26–29]—are used to synthesize the organic semiconductors having proper optical bandgaps and high dipole strengths at their exitation[28,30], which are applied as an interfacial layer of the memristor. To prepare UV-responding molecules, the dinaphthylamine is applied as an electron-donating moiety, and electron-deficient imidazole group is combined to control the highest occupied molecular orbital (HOMO) and lowest unoccupied molecular orbital (LUMO) energy levels of the molecules as D-A molecular structures (DNH, DN, DNH-F, and DN-F in Fig. 1a, b, e, f). DNH and DN were designed in our previous work[31], and DNH-F and DN-F, to which fluorine functional groups are added for the higher polarity (e.g., intrinsic dipole moment), are newly prepared to enhance the photo-responsivity of the device in the UV wavelength region. The asymmetric D-A structure is beneficial for improving intramolecular charge transfer characteristics of the molecule, which is favorable to provide high polarity[29,32], and the dinaphthyl groups in these molecules are also advantageous for extending the conjugation length for high intrinsic dipole moments.

However, the effective conjugation length of the imidazole-based system is not long enough to extend their absorption wavelength to visible region (emission in NIR). Therefore, as visible light-responding molecules, we prepared two NIR-emitting chalcone derivative fluorophores, CH-M and C-M (Fig. 1c, g), in which the hydroxyphenyl part of 2′-hydroxychalcone was replaced by hydroxynaphthyl group and naphthyl group to reduce bandgap of the molecules by introducing nodal-plane model and extending their effective conjugation length. The dimethylamine group is also beneficial to enhancing the D-A strength and further reducing their optical bandgap for covering the entire visible wavelength region by increasing its effective conjugation length. Moreover, to enhance the color-recognition ability of the photonic synapse device, CH-P and C-P (Fig. 1d, h) are further designed with additional modifications. The substitution of electron donating group at 4-position of phenyl group leads to an enhancement of donor–acceptor (D-A) ability of the molecule. Particularly, the dimethyl groups in parent molecules, CH-M and C-M, are replaced by two electron-rich phenyl groups, resulting in the formation of CH-P and C-P, respectively. We will discuss in detail how these molecules enhance the color-recognition ability of the device at a later time.

The general synthetic procedures of the molecules are depicted in "Methods" section and Supplementary Figs. 1 and 2. The energy levels of molecules are obtained by cyclic voltammetry (CV) results (Supplementary Figs. 3 and 4) along with their absorbances (Fig. 1i, j), and their values are added to their CV curves. The detailed information about those molecules (1H NMR, 13C NMR, HR-MS, LC-MS, thermogravimetric analysis, and differential scanning calorimetry) are represented in Supplementary Figs. 5–22 and Supplementary Notes 1–8. UV-Vis spectra of the organic molecules confirm that their absorbances are within UV (DNH, DN, DNH-F, and DN-F in Fig. 1i) or visible region (CH-M, C-M, CH-P, and C-P in Fig. 1j). For reference, the letter H in the

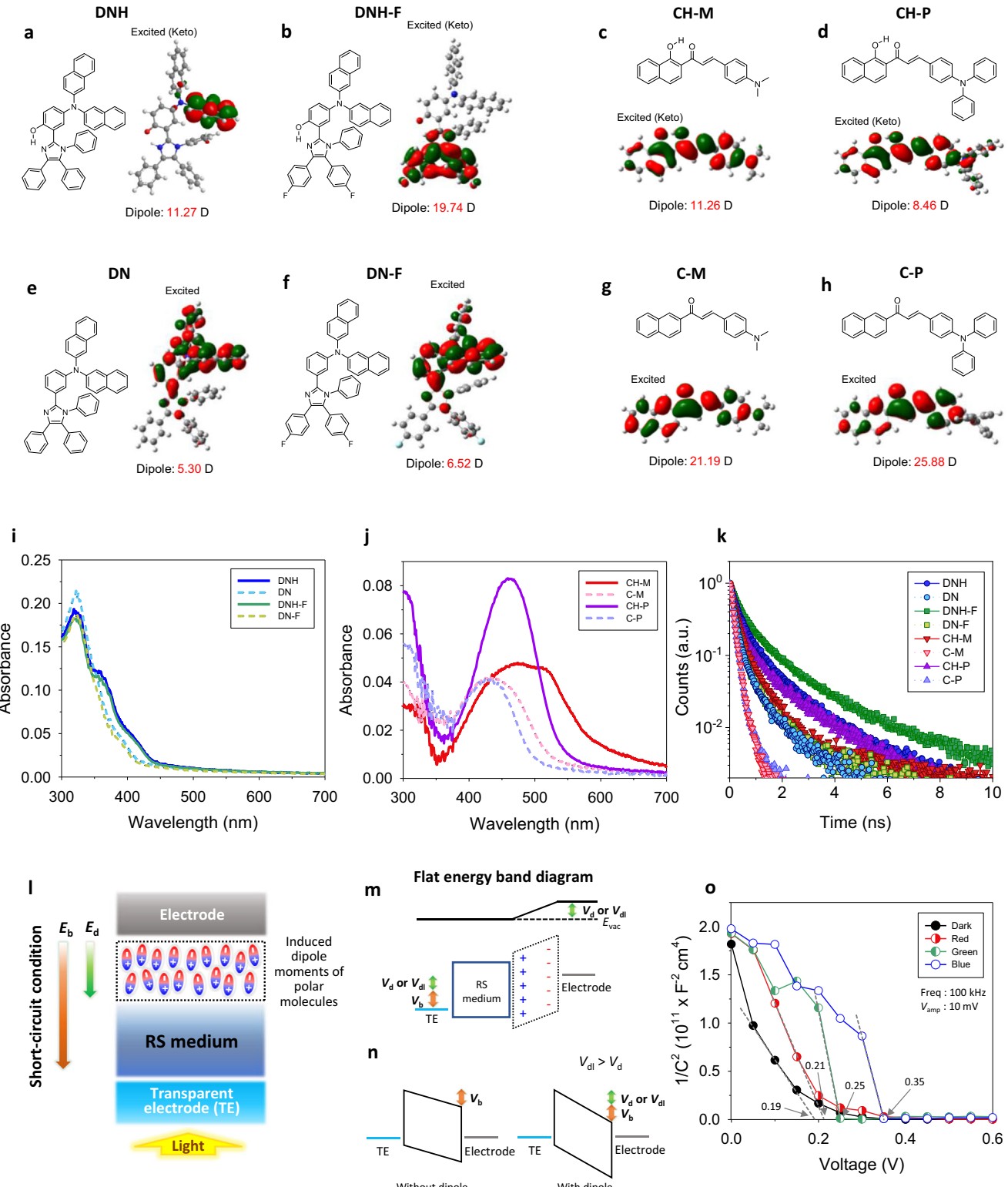

**Fig. 1 | Characteristics of organic semiconductor interlayer enabling optoe-lectronic memristor. a–d** Chemical structures of the ESIPT-active organic semi-conductors and their orbital diagrams in excited states (K* forms) calculated by DFT: **a** DNH, **b** DNH-F, **c** CH-M, and **d** CH-P. **e–h** Chemical structures of the non-ESIPT organic semiconductors and their orbital diagrams in excited state calculated by DFT: **e** DN, **f** DN-F, **g** C-M, and **h** C-P. Theoretically calculated dipole moment values are added. **i,j** Absorbance of the organic semiconductor thin films casted on quartz: **i** DNH, DN, DNH-F, and DN-F. **j** CH-M, C-M, CH-P, and C-P. **k** Time-resolved photoluminescence of organic semiconductor thin films casted on quartz.

**l** Schematic diagram of optoelectronic memristor. **m, n** Energy-level diagrams of the memristor representing the induced dipole moment effect of the molecules within the organic thin film: (m) flat band condition and (n) equilibrium. $V_b$ and $E_b$ are built-in potential and electric field. $V_d$ and $E_d$ are enhanced built-in potential and electric field by the dipole effect. $V_d$ and $E_d$ increase to $V_{dl}$ and $E_{dl}$ at the light ($V_{dl}$ and $E_{dl}$ are enhanced built-in potential and electric field by the dipole effect at the light). **o** Mott–Schottky plots of CH-P-integrated memristor (FTO/NiO/organic interlayer/PMMA/Ag) at dark and under R-, G-, and B-light irradiation (450, 525, and 630 nm).

molecule name indicates an extended lifetime at excitation, which we will explain in the following section.

## Operation of optoelectronic memristor

The delocalized π-electrons in an organic conjugated molecule having a high intrinsic dipole moment are known to provide high polarizability under an external electric field. This is mostly originated from the electronic polarization due to the variation of electron clouds within the molecule by the electric field, fast enough to be approximately $10^{-15}$ s, rather than the rotational polarization, restricted by the lattice in a solid[33–36]. Therefore, if the intrinsic dipole moment of the synthesized molecule is high enough and it is subjected to an electric field, strong induced dipole moment (= polarizability × electric field) is generated, and the accumulative induced dipole moments within a thin film, aligned to the field, are eventually effective. For reference, without the external electric field, the intrinsic dipole moments of molecules within tens of nanometer-thick film are easily canceled out due to their random distribution, although the molecule has high intrinsic dipole moment.

The operation principle of the optoelectronic synaptic memristor, proposed in this work, is represented in Fig. 1l–n. The designed organic semiconductor thin film is inserted between top and bottom electrodes of the device having different work-functions (e.g., FTO-Ag or ITO-Ag), and this potential difference ($V_b$) produces the built-in electric field ($E_b$) within the device (left in Fig. 1n) that can generate induced dipole moments of the molecules within the film (Fig. 1l). If these induced dipole moments aligned to the electric field are strong enough, the accumulative induced dipole moments within the film can change the work-function of neighboring electrode (e.g., decreasing the work-function of Ag as depicted in Fig. 1m), consequently intensifying the built-in potential ($V_b + V_d$) (right in Fig. 1n) and thus electric field within the device ($E_b + E_d$). Given these factors, if these accumulative induced dipole moments within the organic semiconductor interlayer can be additionally enhanced by the light-irradiation, they can further intensify the built-in potential and the internal electric field (from $V_d$ and $E_d$ to $V_{dl}$ and $E_{dl}$: $V_{dl} > V_d$ and $E_{dl} > E_d$)—thus increase the conductance of memristor, which means that the light can control its conductance as an input signal instead of the voltage bias. For reference, the suggested mechanism does not depend on the properties of medium and electrode of memristors.

The electron distribution within an organic molecule is altered at the photo-excited state, and thus its intrinsic dipole moment can be also changed by the light-irradiation. Especially, the multiphenyl-substituted imidazole-based small molecules and chalcone derivative fluorophores have been reported to have the enhanced dipole moment at their excited states[37–40]. Therefore, the light-irradiation on these organic thin films, inserted to the memristor, is expected to enhance the intrinsic dipole moments of the molecules—and thus the accumulative induced dipole moments within the film—that can further intensify the electric field within the device for the conductance increase, as we aim to achieve. It has been reported that the photo-enhanced accumulative induced dipole moments of the polar organic semiconductor interlayer, inserted into the photovoltaic (PV) cell, are strong enough to intensify the built-in potential of the PV device, consequently improving its photo-carrier extraction characteristics and open-circuit voltage[30,31,41]. However, the excited state lifetime of the organic molecule is not generally long enough, and thus its photo-enhanced dipole moment effect at the excitation is not commonly noteworthy in the conventional organic semiconductor.

To make this photo-enhanced dipole effect more valid, we designed our molecules to have photo-tautomerization between enol (E) and keto (K) form at the excitation through intramolecular H-bond (e.g., -OH and -N=), which can follow the excited-state intramolecular proton transfer (ESIPT) process, consequently extending its lifetime at the photo-excitation. The ESIPT process is composed of four-level

cyclic proton-transfer reactions (e.g., E → E* → K* → K → E; * symbol represents the excited state) (Supplementary Fig. 23)[42–44], and this inherent four-level character is due to the stable K* form in the excited state, leading the population inversion in K* state. This inversion extends the effective lifetime of the photo-excited state as K* form, which can make the photo-enhanced dipole moment effect more valid and thus effectively intensify the electric field within the device at the light-irradiation[30,31]. Therefore, light can be utilized as an input signal of the memristor that can replace the electrical voltage bias for the conductance change of the device, even regardless of the type and switching mechanism of the medium. We confirmed through Mott−Schottky analysis, obtained by capacitance-voltage (C−V) measurement (Fig. 1o), that the built-in potential (and therefore its electric field) of the ESIPT thin film-integrated memristor device, represented by $V_b + V_d$, increases (e.g., to $V_b + V_{dl}$) upon exposure to light. The memristor device architecture has the following: FTO/NiO/organic interlayer/PMMA/Ag that will be discussed in detail later again, and $V_{bi} + V_d$ values were calculated by the following equation: $C^{-2} = 2[(V_{bi} + V_d) − V](A^2 e \varepsilon \varepsilon_o N_A)^{-1}$, where $e$ is elementary charge, $\varepsilon \varepsilon_o$ is the permittivity of the component, $A$ is the active device area, and $N_A$ is the doping concentration. The variation of built-in potential value depending on the wavelength of the light will be discussed in detail later again along with the color-cognition functionality of the devices.

DNH (Fig. 1a) is designed to follow the ESIPT process at the UV-light-irradiation, and DN (Fig. 1e) without a hydroxyl group is synthesized as a comparison that cannot follow the ESIPT process to confirm whether the extended lifetime at the photo-excited state by the ESIPT process is crucial for the device to have efficient photo-functionality for the conductance adjustment. In addition, to further prove that the molecule with higher intrinsic dipole moment at the excitation is more effective on the conductance adjustment of the device, DNH-F (Fig. 1b) having fluorine functional groups for the higher polarity is designed, and DN-F (Fig. 1f), of which hydroxyl group is removed to be a non-proton-transfer molecule of DNH-F for comparison, is also prepared. Furthermore, to extend the light-responding functionality of the memristor to the visible region, a chalcone derivative fluorophore, CH-M (Fig. 1c), which can follow the ESIPT process at the visible light-irradiation, is prepared, and C-M (Fig. 1g), a non-ESIPT molecule of CH-M, is also synthesized for comparison. Finally, to optimize the spectral responsivity of the ESIPT molecule for realizing RGB-cognitive photonic memristor, CH-P (Fig. 1d), along with its non-ESIPT molecule C-P (Fig. 1h), are designed.

The dipole moment values of the designed molecules, along with their energy levels and frontier orbital distributions, are calculated by density functional theory (DFT) method. The orbital distributions of molecules at every energy state are displayed with their estimated intrinsic dipole moment values in Supplementary Figs. 24 and 25, and, especially, the orbital distributions and intrinsic dipole moment values at the excited state—K* forms as for the ESIPT molecules (DNH, DNH-F, CH-M, and CH-P)—are represented in Fig. 1a–h. As designed, DNH-F (19.74 D) is confirmed to have much higher dipole moment than DNH (11.27 D) at the excitation. Meanwhile, at the ground state (E form) of CH-P, the molecular orbital mainly exists on electron-rich triphenylamine part of the molecule while that of CH-M is delocalized on entire molecule. Therefore, the substitution of dimethylamine (CH-M) to the strong donor diphenylamine (CH-P) could enhance the intramolecular charge transfer (ICT) characteristics.

Supplementary Figure 26 shows that the photoluminescence (PL) emission spectra of ESIPT molecules (DNH, DNH-F, CH-M, and CH-P) upon $S_0 \to S1$ (ππ*) excitation ($\lambda_{max} = 330$ nm). Especially, DNH and DNH-F, which have similar absorbance spectra with their non-proton-transfer model compounds (DN and DN-F), have much larger Stokes shift than their non-proton-transfer models on the same excitation. This confirms that K* forms with longer wavelength emission are predominantly present at the excitation of DNH and

DNH-F, compared to E* forms. These K* forms in the ESIPT process are proved beneficial for extending their effective lifetimes at the photo-excitation by their PL decay behaviors (Fig. 1k). The time-resolved photoluminescence (TRPL) decay curves in Fig. 1k are convoluted by exponential functions, and it is confirmed that the estimated PL lifetimes of the organic molecules following ESIPT process (DNH: $4.93 \times 10^{-10}$ s, DNH-F: $7.10 \times 10^{-10}$ s, CH-M: $3.16 \times 10^{-10}$ s, and CH-P: $3.89 \times 10^{-10}$ s) are much longer than those of their non-proton-transfer molecules (DN: $2.42 \times 10^{-10}$ s, DN-F: $2.47 \times 10^{-10}$ s, C-M: $1.55 \times 10^{-10}$ s, and C-P: $1.60 \times 10^{-10}$ s) (Supplementary Table 2).

### Induced dipole moment of molecule under electric field

For the operation mechanism of the proposed optoelectronic memristor to be validated, it should be confirmed that the accumulative induced dipole moments within the designed organic thin film are not only notably effective under the electric field, but also further enhanced at the light-irradiation. To evaluate this aspect, we fabricated organic field-effect transistors (OFETs), of which the designed organic molecule thin film was applied as a gate dielectric layer. It has been reported that, if the induced dipole moments within the organic thin film inserted between a semiconductor and a dielectric of OFET can be efficiently aligned to the vertical electric field, they are beneficial for accumulating additional charges in the channel, reducing the threshold voltage ($V_T$) of the OFET devices[45,46]. We fabricated OFETs using 2,7-dioctyl[1]benzothieno[3,2-b][1]benzothiophene (C8-BTBT) and SiO$_2$/DNH(DNH-F, CH-M, CH-P, DN, DN-F, C-M, or C-P)/Al$_2$O$_3$ as a semiconductor and a dielectric layer (Fig. 2a) to assess the accumulative induced dipole moments within the designed organic thin film, and UV and visible light were illuminated to further investigate the photo-enhanced induced dipole moment effect at the excitation. The additional Al$_2$O$_3$ layer on the organic thin film is for excluding the trapping site effect at the interface[45,47].

Figure 2b, c and Supplementary Figure 27a illustrate that the $V_T$ of OFET at dark slightly decreases with an additional organic thin film of polar molecule such as DNH [from −14.0 to −10.9 V (3.1 V ↓: in Fig. 2b and Supplementary Fig. 27a)] and DNH-F [(from −14.0 to −11.8 V (2.2 V ↓: in Fig. 2b, c)] in the dielectric layer (related to the ground-state dipole moments, Supplementary Fig. 24), representing that the accumulative induced dipole moments within the designed organic thin film are effective under the vertical electric field.

More importantly, the $V_T$ values of OFETs with the organic layers further decrease at the light-irradiation (UV 365 nm, 3 mW cm$^{-2}$), and this additional reduction is larger in the OFET with the organic thin film having higher dipole moment at the excitation [from −10.9 to −3.4 V (7.5 V ↓: DNH in Supplementary Fig. 27a) and from −11.8 to −0.8 V (11.0 V ↓: DNH-F in Fig. 2c)]. This proves that, at the light-irradiation, the excited state dipole moments of DNH and DNH-F, higher than their ground-state dipole moments, are effective enough, which is crucial to enhancing the built-in electric field of the memristor by the light as an input signal, and the molecules with higher intrinsic dipole moment at the excitation are more beneficial to generating stronger induced dipole moment under the electric field.

In addition, to evaluate if the extended lifetimes of the molecules at the excitation through the ESIPT process are crucial to the photo-functionality of DNH and DNH-F interlayers, we also prepare OFETs with DN or DN-F as a dielectric layer. Supplementary Figure 27c and Fig. 2e confirm that the $V_T$ variations of the OFETs with DN or DN-F are not noticeable at the light-irradiation compared to those with DNH or DNH-F, indicating that the long-lasting strong photo-enhanced dipoles through the ESIPT process are essential to the operation of the optoelectronic memristors.

Meanwhile, the $V_T$ values of OFETs with the CH-M and CH-P having absorption in the visible wavelength region are confirmed to largely decrease at the visible light-irradiation (blue 450 nm, 5 mW cm$^{-2}$)

(Supplementary Fig. 27b and Fig. 2d); however, similarly, those of OFETs with the C-M and C-P, non-ESIPT molecules, are not noticeable at the same light-illumination condition (Supplementary Fig. 27d and Fig. 2f). For reference, the OFET without the organic dielectric layer also shows slightly decreased $V_T$ at the light illumination (from −14.0 to −12.9/−13.9 V under the same UV and blue-light conditions) (Fig. 2b) because of the photoconductivity effect.

### Memristor device

As a prototype switching medium of the memristor, a NiO thin film, which has been widely utilized as a component of the optoelectronic devices due to its low manufacturing cost and stability[48–50], is applied to our optoelectronic synapse first, and the universality of our approach to various switching medium-based memristors is evaluated later. Especially, a solution-based film growth process is used for the formation of NiO medium because of its easy processability, and, for this purpose, the pre-crystallized solution-processible NiO nanoparticles, which can produce a NP-assembled thin film by the facile spin-casting process, are synthesized. The detailed synthetic procedure for the NiO NP can be found elsewhere[51].

The X-ray diffraction (XRD) patterns of the spin-cast NiO thin film (Fig. 3a) show strong peaks at $2\theta = 37.5°$, $43.3°$, $62.9°$, $75.5°$, and $79.6°$, which correspond to (111), (200), (220), (311), and (222) planes of the cubic crystal structure (JCPDS file no. 65-5745), indicating that the synthesized NPs have a distinct crystalline structure even without any post-process. In addition, we performed a thermal annealing process (550 °C for 2 h) to induce the coalescence of discrete NP crystallites by solid-state necking between adjacent NiO NPs leading to a robust continuous medium. The scanning electron microscopy (SEM) image in Fig. 3b confirms that the grain boundaries (GBs) among those NP crystallites are still observed from the film, indicating that the NP melting does not occur at this temperature. For reference, XRD diffraction peaks of the NiO film annealed at 550 °C for 2 h (Fig. 3a) confirm that the crystal structure of thermally annealed NiO thin film is not noticeably changed from that of the pristine NiO. The resultant polycrystalline medium is composed of three-dimensionally connected GB network that can work as ion-transport channels for the conductive metal filament-based memristor[16]. With this medium, the memristor is designed to have the filamentary redox reaction of active metal cation (e.g., Ag and Cu) for tuning its resistance, which has been known to provide superior analog switching characteristics due to the high mobility of metal ions[52,53].

Supplementary Figure 28 illustrates the cross-section of the optoelectronic memristor device. For the light-illumination onto the designed organic interlayer, the transparent fluorine-doped tin oxide (FTO) electrode and the glass substrate are utilized, and the organic interlayer can be added on the NiO medium due to the superior optical transparency of the NiO. In addition, a PMMA layer is introduced onto the organic interlayer by transfer-printing to improve the uniformity of the switching medium composed of NP agglomerates and to fill the voids within the medium inducing a direct contact between electrodes. This thin polymer layer does not significantly affect the migration behavior of metal cation through the medium[54]. The light-illumination direction, the composing layer stack-sequence, and the film formation process are adjustable depending on the selection of the switching medium and electrodes.

Figure 3d, e are current ($I$)−voltage ($V$) characteristics of the memristor devices, having the following configuration: FTO/NiO/organic interlayer/PMMA/Ag. The resistance of device is changed from high resistance state (HRS) to low resistance state (LRS) by applying positive voltage bias on the top Ag electrode (e.g., FTO is grounded), which represents a set process, and its resistance is recovered to HRS with negative voltage bias as a reset process. This voltage polarity of those set and reset behaviors confirms its bipolar switching property. For reference, the kind of organic interlayer does not affect the $I$−$V$

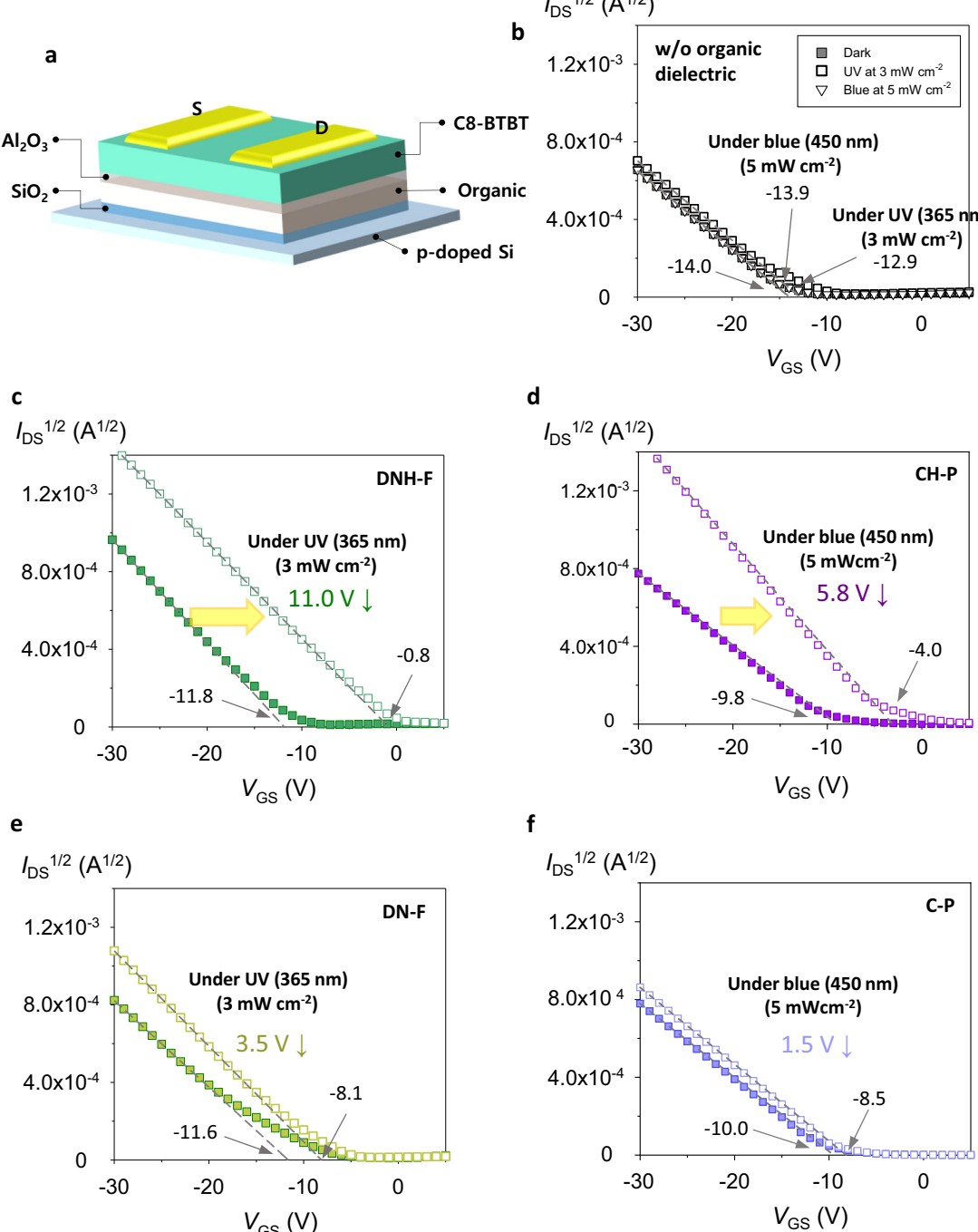

**Fig. 2 | OFET proving induced dipole moment generation within organic interlayer under electric field. a** Schematic diagram of OFET. **b**–**f** Transfer characteristics of OFETs ($I_{DS}$ vs. $V_{GS}$) at $V_{DS}$ = −15 V at dark and light-irradiation condition: **b** without organic dielectric layer, **c**, **d** with ESIPT-active organic thin film as gate dielectric (**c**: DNH-F and **d**: CH-P), and **e**, **f** with non-ESIPT organic thin film as gate dielectric (**e**: DN-F, and **f**: C-P).

characteristics of the device (Fig. 3d, e). To prevent the permanent breakdown of the device, the compliance current ($I_{cc}$) is set to be 15 mA, and $I$–$V$ characteristics of the devices are measured with the following bias condition, 0 → 2 → −2 → 0 V, as one cycle. Figure 3d, e are 10-cycle d.c. $I$–$V$ characteristics of the memristors after electro-forming, and their 1st cycle plots for the electro-forming are represented in Supplementary Fig. 29 for reference.

The temporal cycle-to-cycle variation (CCV) results of the organic interlayer-integrated memristor confirm that it does not show any noticeable degradation of HRS-LRS ratio (Fig. 3g), $I$–$V$ characteristics (Fig. 3h), and set voltage (Fig. 3i) even after $1.1 \times 10^5$ sweeping cycles,

representing the robustness of the organic interlayer. Furthermore, the organic layer-integrated memristor preserves its conductance values at LRS and HRS (DNH-F: 0.32% and 0.03%, CH-P: 1.40% and 0.02% relative standard deviation values [standard deviation($\sigma$)/mean($\mu$)] at LRS and HRS over 230 h), representing its durability (Fig. 3j). In addition, Supplementary Fig. 30a illustrates the retention behaviors of the device measured at various temperatures, fitted to an Arrhenius plot[16], indicating that it can withstand more than 10 years at room temperature without any significant degradation in its performance. We also conducted D.C $I$–$V$ sweep cycles (Supplementary Fig. 30b) and evaluated the light responsivity of the device

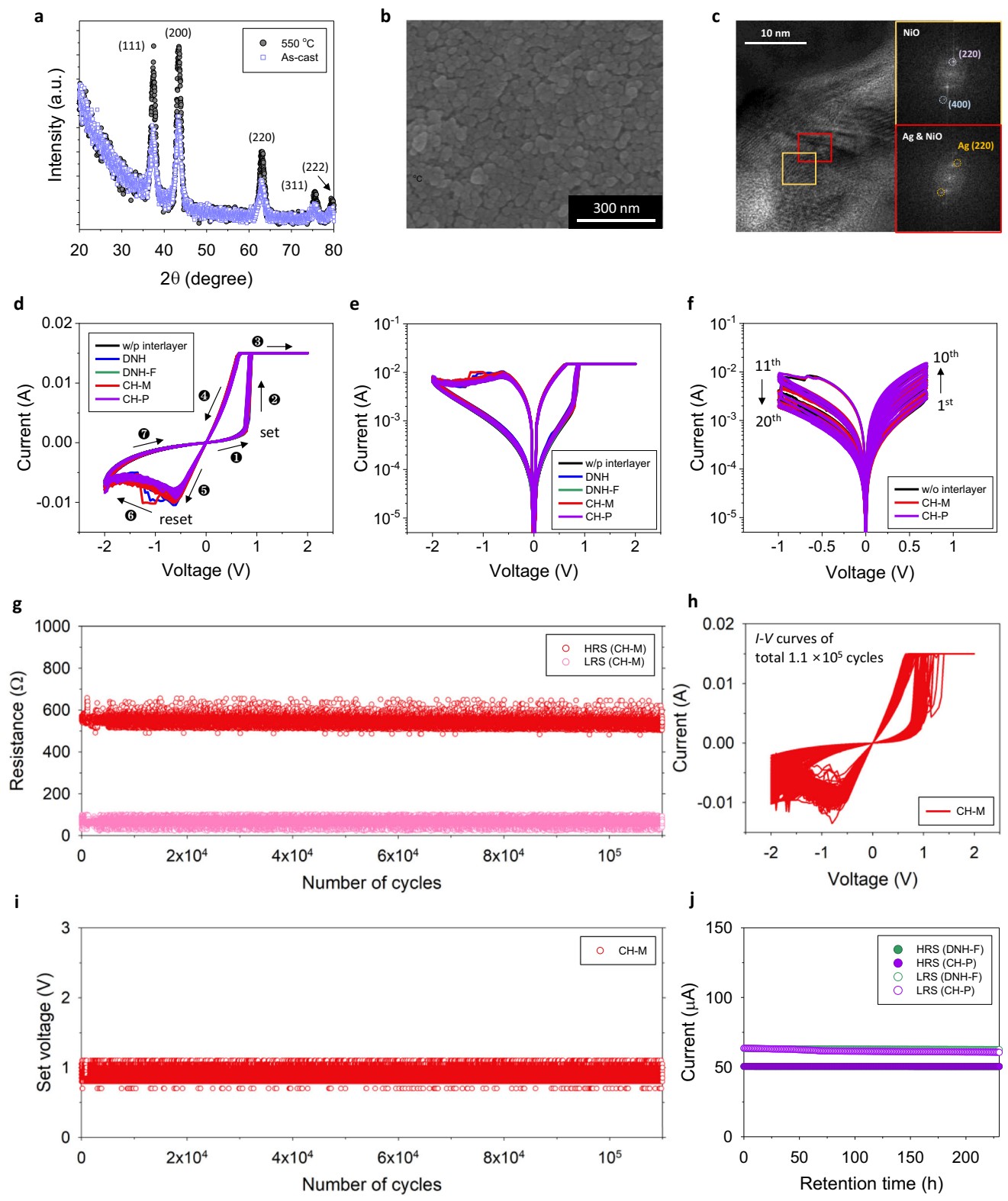

(Supplementary Fig. 30c), even after these high temperature retention tests, and ensured that it maintained similar performance and trends as before the tests.

Meanwhile, for the device to be utilized as a synaptic element, the current level of the memristor should gradually change with continuous voltage sweeps. Figure 3f confirms that the designed memristor exhibits analog gradual set and reset behaviors by the positive and negative voltage sweeps on the Ag electrode, and these

bidirectional analog switching properties correspond to the potentiation and depression characteristics of the artificial synapse.

Figure 3c is a cross-sectional high resolution-transmission electron microscopy (HR-TEM) image of the NiO medium in LRS state. The NiO cubic crystal structure is confirmed by the FFT patterns of the selected area (yellow-square), and the additional Ag crystal structure together with the NiO crystal structure is also observed (red-square), representing the migration of Ag ions through the NiO switching

**Fig. 3 | Memristor device characteristics. a** XRD patterns of NiO nanoparticle-based thin film before and after thermal annealing at 550 °C for 2 h. **b** SEM images of NiO thin films on FTO after thermal annealing at 550 °C for 2 h. **c** HR-TEM image of NiO medium of the memristor in low resistance state (LRS). Upper right and lower right insets are fast Fourier transform (FFT) images of the selected areas in (**c**) (yellow and red regions). **d**, **e** Measured 10-cycle d.c. *I–V* characteristics of the memristor after electro-forming built with various organic thin films. Positive voltage bias is first applied to top Ag electrode (FTO: grounded): **d** normal scale (numbers represent the order of the applied voltage). **e** log scale. **f** Analog multi-level resistive switching properties of the memristor on a log scale. 10 consecutive positive voltage sweeps on Ag followed by 10 consecutive negative voltage sweeps. **g**, **i** Temporal cycle-to-cycle variation (CCV) of (**g**) the resistance of the CH-M-integrated memristor in HRS and LRS (read voltage: 0.05 V) and (**i**) the set voltage, obtained by *I–V* characteristics (after electro-forming). **h** Measured d.c. *I–V* plots of CH-M-integrated memristor during the CCV test for total $1.1 \times 10^5$ cycles (1 plot per 100 cycles is represented). **j** Retention test results of DNH-F and CH-P-integrated memristors at LRS [after 50 consecutive UV pulses (365 nm, power: 3 mW cm$^{-2}$, width: 0.5 s, and interval: 0.5 s) and B-light pulses (450 nm, power: 5 mW cm$^{-2}$, width: 1 s, and interval: 0.5 s)] and HRS. Read voltage 0.05 V is applied during test.

medium. In addition, we prepared a memristor device with inactive Au top electrode on the NiO medium, but we could not find any bipolar switching behavior from this device (Supplementary Fig. 31), indicating that the migration of active metal cation was crucial to the operation of the NiO-based memristor.

## Synaptic properties

To demonstrate a neuromorphic visual system, the color-cognitive synaptic devices that can discriminate and memorize the light signals are required. In this section, we evaluate the photosensitive synaptic characteristics of the designed memristors with DNH and DNH-F at the UV-light pulse conditions, and the indispensability of ESIPT process, which can provide the long-lasting strong excited state dipole moment effects, is confirmed by comparing their synaptic behaviors with those of memristors with the non-proton-transfer molecules, DN and DN-F. Moreover, the visible color-distinguishable synaptic characteristics of the memristors with CH-M and CH-P are further investigated.

Supplementary Figure 32a, c illustrate the excitatory postsynaptic current (EPSC) response of the memristor with the DNH interlayer at an UV-light pulse (365 nm) as a presynaptic signal. The postsynaptic current (PSC) of the device is suddenly increased with the light pulse, and its height is intensified with the amplitude (Supplementary Fig. 32a) and width (Supplementary Fig. 32c) of the light pulse, representing that the light pulse can be used as an input signal replacing the voltage pulse. Although the detailed photophysical dynamics for the density of exciton in this organic interlayer at the light-illumination conditions needs further studies, we believe that it is large enough because of the high absorption coefficient of the molecule. Therefore, at these light-illumination conditions, both ground and excited state molecules coexist in the DNH thin film, but considerably large number of excited molecules having higher dipole moments exist, as they survive efficiently due to the increased lifetime at the excitation by the ESIPT process. The indispensability of this extended lifetime at the excitation is further confirmed by the EPSC response of the memristor with the non-ESIPT DN interlayer at the light-illumination, which is negligible (Supplementary Fig. 33a). This trend is well-matched to the performance variation of OFETs at the light-irradiation (Supplementary Fig. 27a, c), explained earlier. Consequently, the accumulative induced dipole moments of the DNH molecules within the film, which are stronger at their excitation, are effective enough to enhance the electric field within the memristor at the light-illumination, and they help us to use the light pulse as input signal instead of the voltage pulse. Meanwhile, as the light power irradiated on the DNH thin film increases (Supplementary Fig. 32a, c), the density of the excited molecule increases, and thus irradiating higher light power on the device becomes equivalent to applying higher voltage bias to the memristor. This feature can be a basis of the color-distinguishable synaptic memristor, which will be discussed later in detail.

We also confirm that the PSC response of the memristor is further intensified with the consecutive two light pulses (Supplementary Fig. 32e), and this represents the paired-pulse facilitation (PPF) characteristics of the synapse. The paired-pulse facilitation (PPF) index is defined by the ratio of the second PSC spike ($A_2$) to that of the first PSC spike ($A_1$) and it is exponentially increases as the interval between those light pulses ($\Delta t$) decreases[16], while the PSC rapidly drops after the pulse, which corresponds to the short-term plasticity of the biological synapse. This PPF characteristics can be fitted by the exponential function: PPF index $= C_1 \exp(-\Delta t/\tau_1) + C_2 \exp(-\Delta t/\tau_2) + 1$, where $\tau_1$ and $\tau_2$ are rapid and slow phase relaxation times, and $C_1$ and $C_2$ are their initial facilitation. $\tau_1$ and $\tau_2$ values of the memristor with various organic semiconductors are summarized in Supplementary Table 3.

We have shown that the molecule with higher intrinsic dipole moment at the excitation is more advantageous for providing stronger induced dipole moment under the electric field at the light-illumination (Supplementary Fig. 27a and Fig. 2c). Therefore, the memristor with an organic interlayer having high intrinsic dipole moment at the excitation is also expected to be beneficial for enhancing its photo-responsivity. The EPSC and PPF behaviors in Fig. 4a, c, e show that the memristor with DNH-F having higher dipole moment exhibits stronger synaptic photo-responsivity than that of the memristor with DNH at the same UV-light pulse conditions (amplitude, width, and interval) (Supplementary Fig. 32a, c, e). For reference, the memristor with DN-F, non-proton-transfer molecule of DNH-F, also does not efficiently respond to the light pulse signal (Supplementary Fig. 33b), similar to that of the memristor with DN (Supplementary Fig. 33a), indicating the importance of the long-lasting strong photo-enhanced dipoles through the ESIPT process, again.

The PPF behavior of the memristor represents the short-term memory of the synapse, of which the PSC rapidly decays without the input signal, but it also accompanies the increase in conductance even after the pulse as a non-volatile memory characteristic (insets of Supplementary Fig. 32e and Fig. 4e). This conductance change, sustaining the PSC at a certain level after the pulse, is observed even after a single pulse signal, if the pulse is high (Supplementary Fig. 32a and Fig. 4a) and large (Supplementary Fig. 32c and Fig. 4c) enough. This non-volatile conductance variation, representing the weight adjustment of synapse, can be a basis of the conversion of short-term memory to long-term memory, providing long-term plasticity to the memristors. Supplementary Figure 34a and Fig. 5a illustrate the gradual increase of PSC spike at the consecutive UV-light pulse train (365 nm), and the enhanced conductance of the memristor after the pulse train is also observed. This updated conductance of the memristor can be refreshed by the electrical voltage pulses (e.g., negative voltage bias on the top Ag electrode with the bottom electrode grounded), and the consequent long-term potentiation (LTP) and depression (LTD) characteristics of the memristor depending on the pulse number, representing the long-term plasticity of the artificial synapses, are illustrated in Supplementary Fig. 34c and Fig. 5c. Meanwhile, Supplementary Fig. 34a, c and Fig. 5a, c confirm that the memristor with the DNH-F, which exhibits stronger synaptic photo-responsivity than that of the memristor with DNH due to its higher dipole effect at the excitation (Supplementary Fig. 32a, c, e and Fig. 4a, c, e), has larger dynamic range, as expected.

To prove that the proposed operation principle is applicable to various memristors regardless of the type and switching mechanism of the medium, we prepared the DNH-F-integrated memristors built with

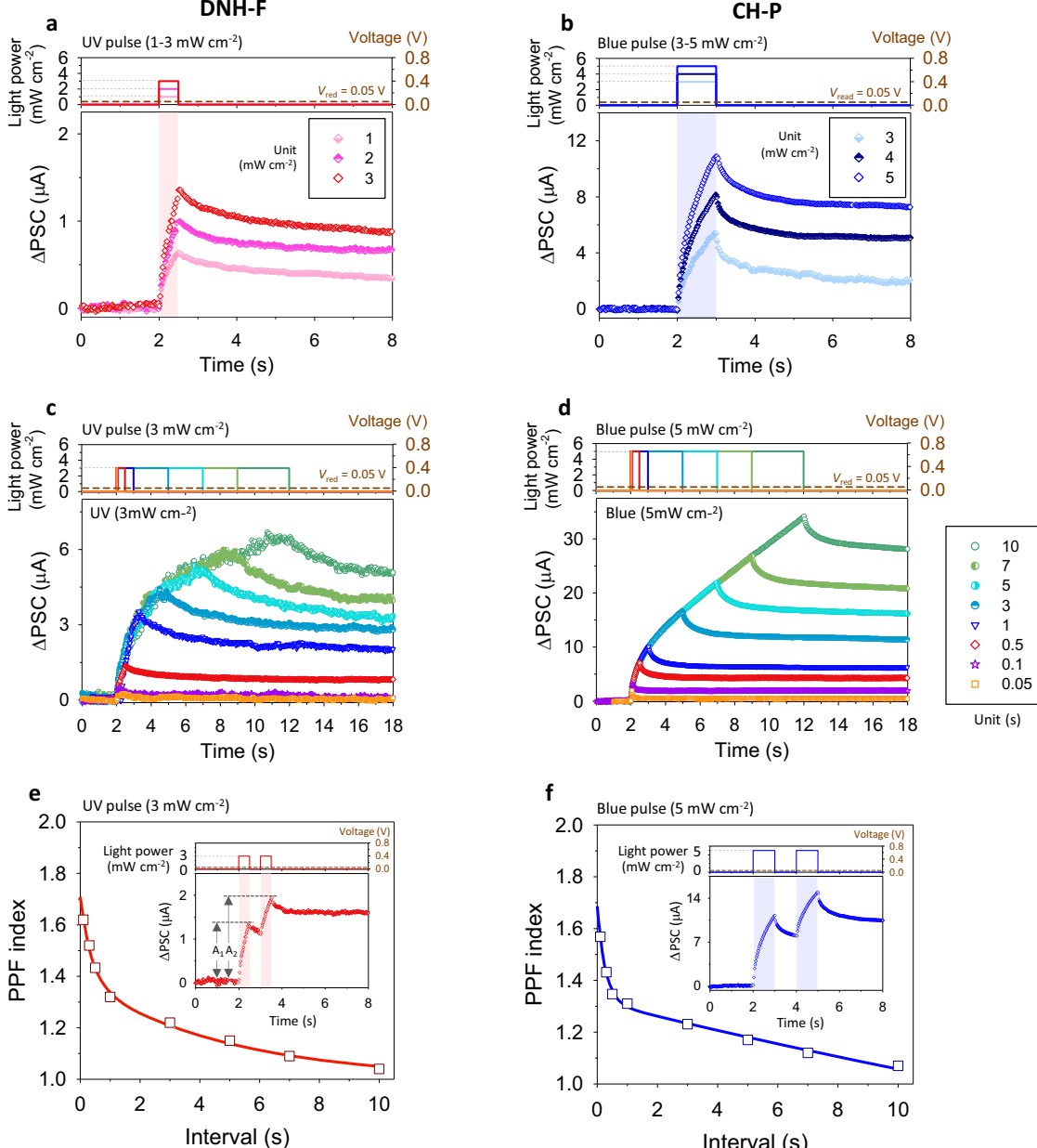

**Fig. 4 | Optoelectronic synapse characteristics. a–d** EPSC responses of the memristors integrated with various organic thin films: **a, c** DNH-F and **b, d** CH-P. Light pulse conditions are illustrated in top figures. Read voltage is 0.05 V. **a, b** Depending on the amplitude of the light pulse (pulse widths are fixed): UV pulse (365 nm and 0.5 s width) and B-light pulse (450 nm and 1 s width) are used for the memristor with DNH-F and that with CH-P. **c, d** Depending on the width of the light pulse (pulse amplitudes are fixed): UV pulse (365 nm and 3 mW cm$^{-2}$) and B-light pulse (450 nm and 5 mW cm$^{-2}$) are used for the memristor with DNH-F and that with CH-P. **e, f** PPF index as a function of interval between two consecutive pulses (Δ$t$). Read voltage is 0.05 V. **e** Amplitude and width of UV pulse (365 nm) for the memristor with DNH-F are 3 mW cm$^{-2}$ and 0.5 s. **f** Those of B-light pulse (450 nm) for the memristor with CH-P are 5 mW cm$^{-2}$ and 1 s. Bottom inset figures of (**e, f**) are about PPF behaviors by a pair of presynaptic pulses for 0.5 s (DNH-F) and 1 s (CH-P) interval cases. Top inset figures of (**e, f**) illustrate the applied light pulse conditions.

various switching mediums (Al$_2$O$_3$, FeO$_x$, CuO$_x$, SnO$_2$, and PMMA)[55]. Figure 5e shows LTP/LTD characteristics of these memristors, and all of them are confirmed to work successfully as optoelectronic synapses that can efficiently respond to the consecutive light pulses. In addition, Cu top electrode is applied on the DNH-F-integrated memristors instead of Ag, and its LTD/LTP characteristics in Fig. 5f confirmed that our approach is not also limited to the specific electrode.

In the human visual system, the colors are distinguished by the cone photoreceptor cells in the sensory membrane, retina, and first-stage pre-processing is also performed here through the synapses connecting the hierarchical neurons. This pre-processed data is then transferred to the visual cortex of the brain through the optic nerve for complex high-level processing. The color-distinguishable synaptic memristor, which can emulate the function of the human retina, can be prepared by integrating the visible light-sensitive CH-M and CH-P interlayer into the memristor device. Supplementary Figure 32b, d, f illustrates that the CH-M-integrated memristor can efficiently respond to the blue-light pulse (450 nm) as a synaptic device, showing the EPSC and PPF behaviors. It can also efficiently respond to the consecutive visible light pulse train (450 nm), which shows the gradual increase of PSC spike along with the conductance update after the pulse train (Supplementary Fig. 34b). The consequent LTP/LTD characteristics of

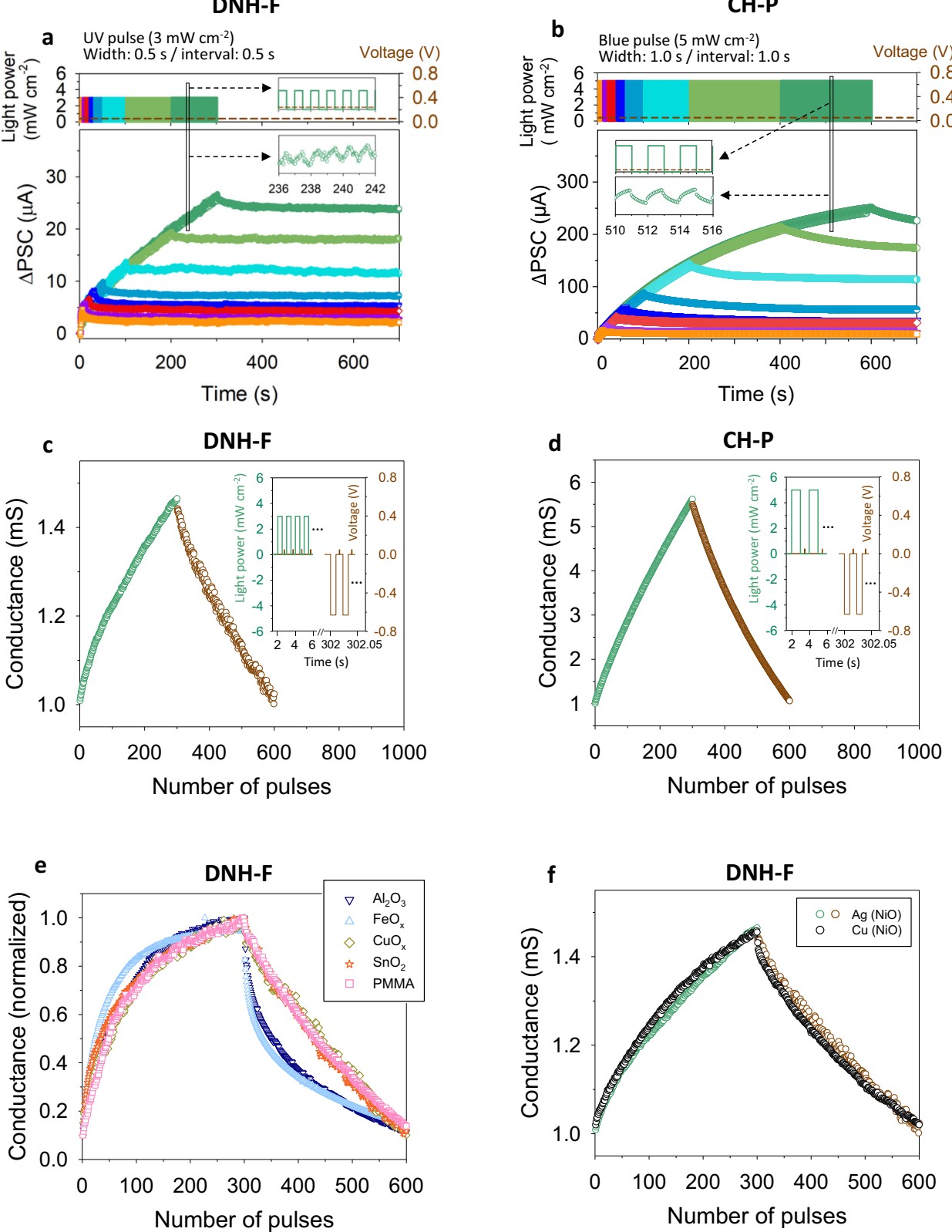

the CH-M-integrated memristor are represented in Supplementary Fig. 34d. For reference, similar to the memristors with DN and DN-F, non-proton-transfer molecules, the C-M-integrated memristor does not efficiently respond to the light pulse signal (Supplementary Fig. 33c), representing the importance of the extended lifetime at the excitation through the ESIPT process again.

The strong dipole effect of organic molecules at the excitation is more effective when the density of the excited molecules increases, which is determined by the density of the absorbed photon and their lifetime at the excitation. As explained, we could extend the lifetime of the excited molecule using the stable K* form in the ESIPT process. In general, the excited state lifetime of an organic semiconductor

**Fig. 5 | Optoelectronic synapse characteristics at consecutive light pulse train.** **a**, **b** Pulse number-dependent PSC: **a** DNH-F and **b** CH-P. In inset figures, *x*-axis about time is enlarged. Light pulse conditions are depicted in top figures. UV pulse (365 nm, power: 3 mW cm⁻², width: 0.5 s, and interval: 0.5 s) and B-light pulse (450 nm, power: 5 mW cm⁻², width: 1 s, and interval: 1 s) are used for the memristor with DNH-F and that with CH-P. Read voltage is 0.05 V. **c**, **d** LTP/LTD characteristics: **c** DNH-F and **d** CH-P. Light and voltage pulse conditions for LTP and LTD are depicted in inset figures. Pulse conditions of potentiation(P)/depression(D) of the

memristor with DNH-F are 365 nm, 3 mW cm⁻² power, 0.5 s width, and 0.5 s interval/−0.63 V, 0.01 s width, and 0.013 s interval. Those of the memristor with CH-P are 450 nm, 5 mW cm⁻² power, 1 s width, and 1 s interval/−0.63 V, 0.01 s width, and 0.013 s interval. Amplitude and width of the read voltage pulse are 0.05 V and 0.001 s. **e**, **f** LTP/LTD characteristics of the DNH-F-integrated memristors depending on various **e** switching medium and **f** electrode. The same pulse conditions as those in (**c**) are used for P/D and read.

---

undergoing ESIPT can exhibit a trend where it increases with decreasing wavelength (or increasing energy) of the incident light[56–58]. This is because higher-energy photons (shorter wavelength light) can provide the necessary energy to overcome the barrier for efficient and rapid proton transfer processes from E* to K*, leading to a stabilization of the excited state and a longer excited state lifetime–a more detailed discussion regarding this can be found in Supplementary Note 9. Supplementary Figure 35a confirms that the PL lifetime of CH-M excited at shorter wavelength is much longer than that excited at longer wavelength. Therefore, as for the memristor built with the CH-M thin film, which has higher optical absorption and longer PL lifetime at the shorter wavelength in visible region, the photo-enhanced dipole effect is more prominent at the light-irradiation in the shorter wavelength region, and this provides the different degree of PSC rise and conductance update depending on the wavelength of the incident light. Supplementary Figure 36a illustrates the EPSC responses of the CH-M-integrated memristor, exposed to red (R, λ = 630 nm), green (G, λ = 525 nm), or blue (B, λ = 450 nm) light pulse at the same power condition (5 mW cm⁻²), and it exhibits large wavelength-dependent PSC spike variation and conductance change after the pulse, indicating distinct color-distinguishable photo-synaptic characteristics. This color distinction becomes more noticeable as the number of light pulses increases, and the signals do not overlap even after 50 consecutive light pulses (Supplementary Fig. 36b). The variation range of PSC output (and the related conductance) during those light pulses is separated at each color and preserved even after the pulses, representing the color-recognition capability of the memristor (Supplementary Fig. 36c). However, this range is not distinctly separated at each color. As the pulse number increases, the maximum PSC values of the devices under R- and G-light pulses approach the minimum PSC values under G- and B-light pulses, respectively. Therefore, this does not guarantee color discrimination under different light power conditions.

CH-P is designed to enhance the photo-synaptic characteristics and color-discrimination ability of the device, even under different light power conditions. This is achieved by increasing its absorbance around 450 nm, resulting in a sharper absorbance curve that covers the visible wavelength range. As a result, the absorbance difference between 450 and 650 nm, which are the wavelengths targeted for discriminating RGB colors (Fig. 1j), is enhanced. Moreover, its PL lifetime increases as the excited wavelength decreases (Supplementary Fig. 35b), further magnifying the gradient of the excited state dipole effect under each colored light. The Mott-Schottky plot (Fig. 1o) confirms a change in the built-in potential depending on the wavelength of the exposed light, with the highest and lowest values observed under B and R light, respectively.

Figure 4b, d, f demonstrates the EPSC and PPF behaviors of the CH-P-integrated memristor. At the same light pulse condition (B-light pulse, 450 nm), it responds more efficiently than CH-M-integrated memristor (Supplementary Fig. 32b, d, f). Therefore, it can respond more efficiently to consecutive visible light pulse train (450 nm) (Fig. 5b), and its dynamic range for LTP/LTD characteristics (Fig. 5d) is much higher than that of the CH-M-integrated memristor (Supplementary Fig. 34d). The estimated nonlinearity (β) and symmetricity values of the LTP/LTD characteristics[16], representing the superior

conductance tunability of the prepared CH-P-integrated memristors, are summarized in Supplementary Table 1.

Figure 6a depicts the EPSC responses of the CH-P-integrated memristor exposed to R-, G-, or B-light pulse at the same power condition (5 mW cm⁻²). They exhibit significant wavelength-dependent PSC spike variation and conductance change after the pulse. These RGB-color-dependent photo-synaptic behaviors are amplified with pulse number (Fig. 6b), and we confirm that the output current values of the device under RGB colors, with various intensities ranging from 1 to 20 mW cm⁻², a light intensity range applicable to human visual systems without safety concerns[13,59,60], can be perfectly distinguished (Fig. 6c, d, f). For each RGB-light, the output current values under these various light intensities are in the range of 0.01–0.12 μA, 1.68–4.19 μA, and 104.05–210.37 μA, respectively (Fig. 6d, f), which provide dynamic range values for color-distinction of 28–48 dB, 71–78 dB, and 106–112 dB, respectively (Fig. 6f), representing distinct color-distinguishable photo-synaptic characteristics. The dynamic range for color-distinction is defined by the following equation: dynamic range = $20 \log(\text{PSC}_{max}/\text{PSC}_{min})$, where $\text{PSC}_{max}$ is steady state PSC after 50th pulse under 20 mW cm⁻² at each color, and $\text{PSC}_{min}$ is steady state PSC after 1st pulse under 1 mW cm⁻² at R-light pulse.

## Emulation of human visual system

The designed optoelectronic memristors can be building blocks to achieve an artificial visual system recognizing colorful images with high accuracy. The memristor array, composed of the CH-P-integrated optoelectronic memristors having color-distinguishable synaptic characteristics, is used to detect and pre-process the colorful input signals, as the retina performs in the human eye, and the CNN, composed of the DNH-F-integrated optoelectronic memristors, is applied for performing the complex image training and recognition tasks, similar to the high-level processing of the visual cortex in the human brain.

As a proof of concept, a photonic synapse array with 8 × 8 pixels is prepared (Fig. 6e), and its sensing and pre-processing results about an RGB-colored letter pattern image input are illustrated in Fig. 6g. Through the memristor array, the exemplary light patterns, which consist of three different colored letters (red-colored "H", green-colored "U", and blue-colored "i") but have the same pulse conditions regardless of the color (power: 5 mW cm⁻², width: 1 s, and interval: 0.5 s), can be classified into the three different letters, represented by the assembly of the memristors having the distinct range of conductance—and the related PSC—depending on the color. Moreover, these pre-processed output signals are preserved within their distinct distributions even after the pulse train as a non-volatile long-term memory, which is correlated to their superior retention characteristics (Fig. 3j).

To confirm the color image processability of the artificial visual system, the image recognition tasks about the Canadian-Institute-For-Advanced-Research-10 (CIFAR-10) dataset, which includes 10 colorful object categories, composed of 50,000 images for training and 10,000 images for inference, are performed. Every color input image is separated into RGB three channels through the CH-P-integrated memristor-array as a pre-processing, and these are introduced into the eight-layered CNN, composed of three convolution layers with three pooling

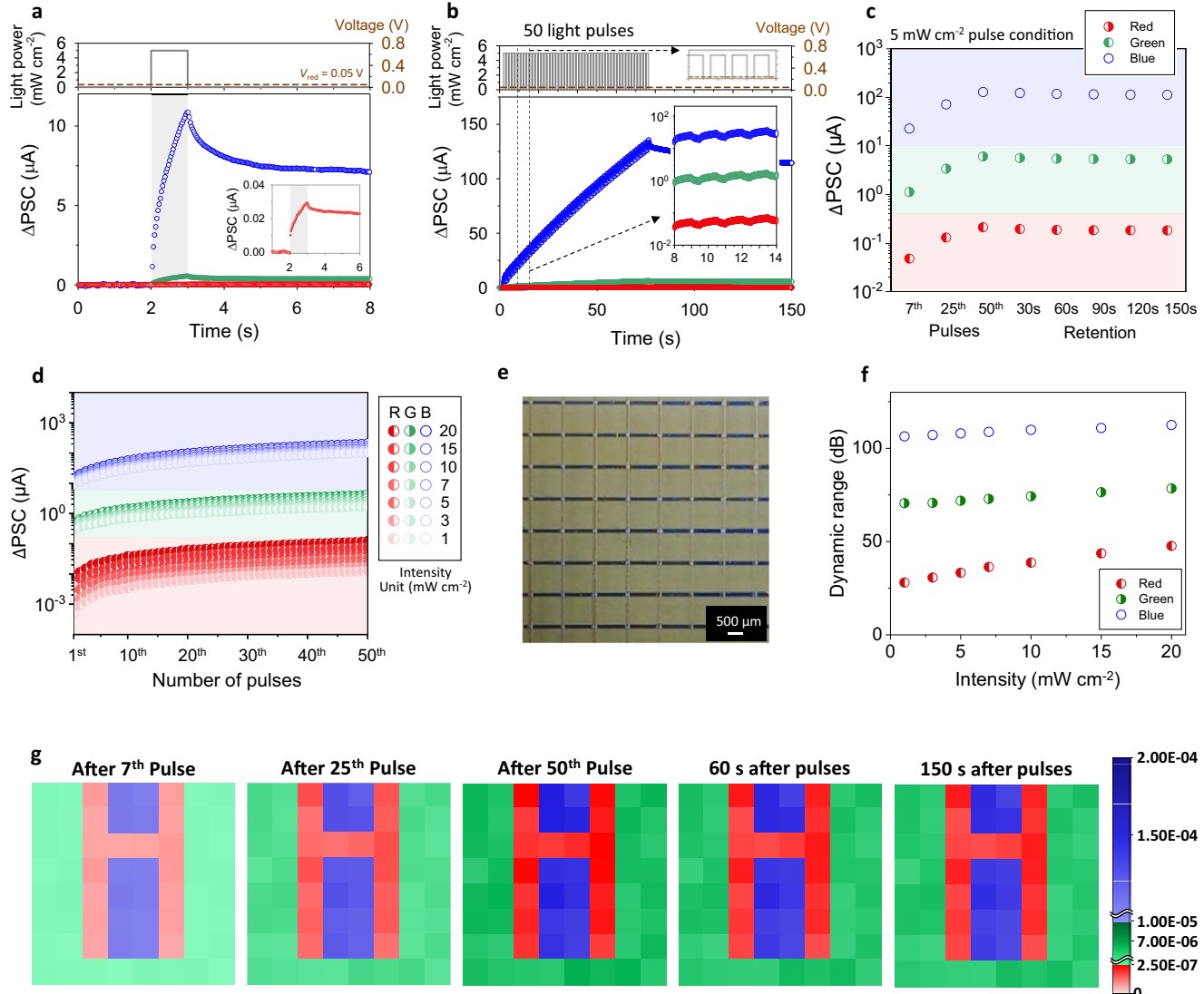

**Fig. 6 | Optoelectronic synapse for RGB multispectral color-discrimination.**
**a** EPSC responses of CH-P-integrated memristor under B-, G-, and R-light pulse irradiation (450, 525, and 630 nm) having the same pulse condition (5 mW cm⁻² power and 1 s width). Light pulse condition is depicted in top figure. Read voltage is 0.05 V. **b** PSC variations of CH-P-integrated memristor under 50 consecutive pulses of B-, G-, and R-light (450, 525, and 630 nm) having the same pulse conditions (5 mW cm⁻² power, 1 s width, and 0.5 s interval). Light pulse condition is depicted in top figure. Read voltage is 0.05 V. In inset figures, x-axis about time is enlarged. **c** PSC distribution depending on the number of pulses and retention time, obtained from 8 × 8 pixels of CH-P-integrated memristor array, for color discrimination. RGB

pulse conditions are the same with (**b**). Circles and error bars represent average and standard deviation. **d** PSC distribution depending on the number of pulses, obtained from CH-P-integrated memristor under RGB-light pulses with various light intensities from 1 to 20 mW cm⁻² (RGB pulse conditions: 0.5 s width, and 2 s interval). **e** Optical microscopy image of 8 × 8 pixels of CH-P-integrated memristor array. **f** Dynamic range of CH-P-integrated memristor for color-distinction under various light intensities from 1 to 20 mW cm⁻². RGB pulse conditions are the same with (**d**). **g** 2D contour maps of PSCs measured from 8 × 8 pixels of CH-P-integrated memristor array at different pulse and retention time conditions. RGB pulse conditions are the same with (**b**).

layers for the feature extraction and two fully-connected-layers (FCLs) for the classification (Fig. 7a). The hardware memristor-crossbar arrays for the two FCLs are depicted in Fig. 7d. The synaptic weight ($W$) of each cell is defined by the conductance difference between two equivalent DNH-F devices ($W = G^+ - G^-$) to represent negative weights as well as positive values with the conductance of the memristors that can only have positive values[61,62].

As a reference, the pre-processing of CIFAR-10 colorful images, composed of pixels to which RGB-mixed color is illuminated, involves separating the RGB mixed color into three individual RGB colors (Supplementary Fig. 37). This is different from the case where individual RGB monochromatic light is illuminated to each pixel (Fig. 6g). The CH-P-integrated memristor possesses the capability to discriminate an RGB-mixed color into 4-bit resolution individual RGB

colors [e.g., (R, G, B) = (0–15, 0–15, 0–15)], based on its characteristics, such as the strength of the excited state dipole moment, spectral responsivity, and spectral excited state lifetime. Further details about the resolution of the color-discrimination capability are provided in Supplementary Note 10. By designing molecules with enhanced excited state dipole moment, along with improved spectral responsivity and spectral excited state lifetime, we can further increase the individual color resolution by optimizing the conductance gap between each state of each color.

By composing the convolutional kernels and FCLs with the DNH-F-integrated memristor arrays having the strongest photo-synaptic functionalities, a light-based intelligent system, of which synaptic weights can be updated by the optical stimulation, can be proposed. The conventional electrical programming offers a reliable means of

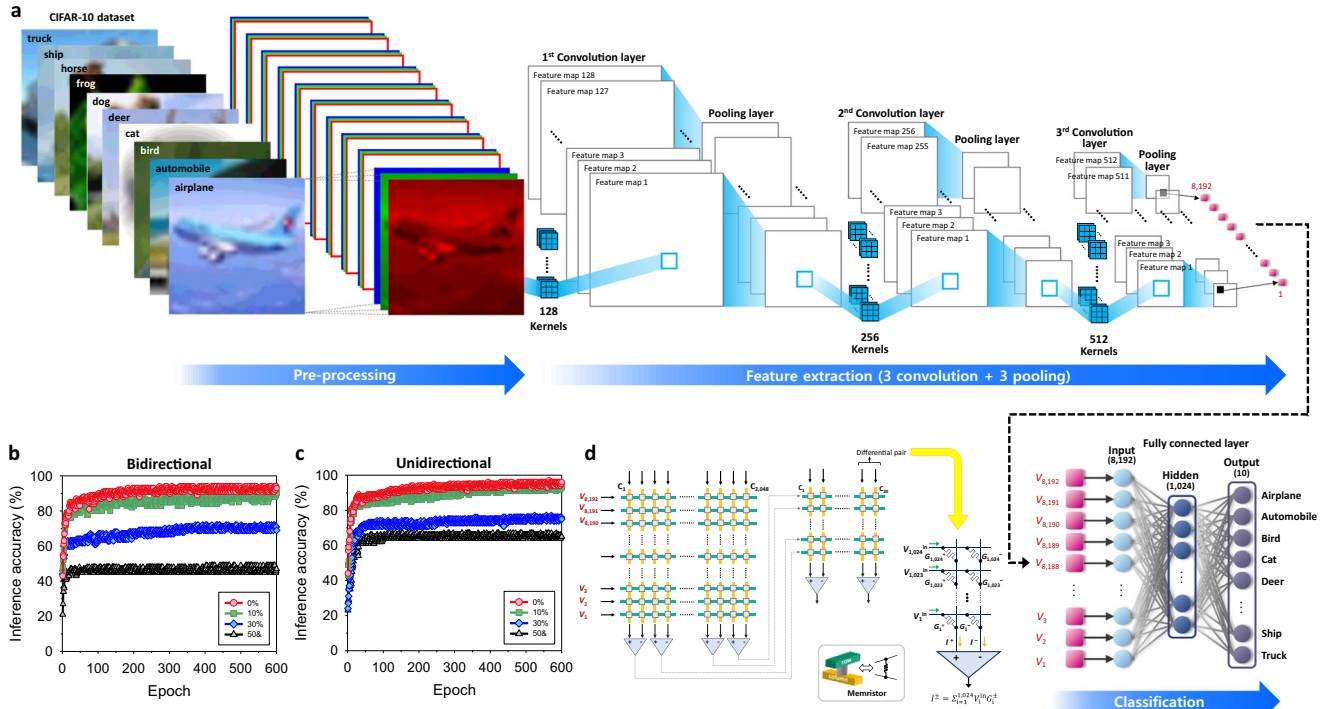

**Fig. 7 | Optoelectronic synapses for convolutional neural network.**
**a** Convolutional neural network (CNN), composed of convolutional, pooling, and fully connected layers, about the CIFAR-10 dataset, pre-processed for RGB color discrimination. **b**, **c** Image recognition rate of the CIFAR-10 dataset: **b** bidirectional and **c** unidirectional weight update. **d** Memristor crossbar arrays for the fully connected layers of the CNN.

quantization to represent multiple bits for memristors. For instance, recently, memristor crossbars have achieved up to 2048 conductance states using electrical programming protocols[63]. However, the optical neural network system provides additional advantages, allowing for higher spatiotemporal resolution and simplification of the weight-update methodology in the hardware array[20]. Moreover, optical programming holds promise for ultrafast speeds, low energy consumption, and large bandwidths by utilizing photons[64]. The utilization of light-irradiation as an additional non-contacting terminal for data-writing can reduce power loss from electrical interconnects during the electrical programming[65]. Based on this, the unidirectional method, which utilizes light-only signals, and bidirectional method, which applies light and voltage signals for potentiation and depression, can be used for updating the synaptic weights[16,20,66]. The training and inference tasks of the CNN to the colorful CIFAR-10 dataset images after their pre-process are simulated by the DNN+ NeuroSim framework[67] with Python, and the detailed information of the feed-forward and backpropagation procedures of the CNN including the models for the weight update are explained in "Methods" section and Supplementary Note 11.

Figure 7b, c shows that the inference accuracy of the CNN, composed of the designed optoelectronic memristors (4-bit RGB image resolution), about the CIFAR-10 dataset approaches to 94% (bidirectional) and 96% (unidirectional) after hundreds of epochs with the pre-processed input signals having distinct discrimination among RGB colors. In contrast, if the RGB characteristics composing the colorful input signals are not efficiently distinguished into their pure components for the three individual channels (Supplementary Fig. 37), those inference accuracies of the network are largely decreased and even degraded to 49% (bidirectional) and 67% (unidirectional), when the overlapped RGB components reach 50%. This clearly demonstrates that the distinct color discrimination through the pre-process is crucial to achieving high recognition rate from the artificial visual system. For reference, when the optoelectronic memristors with a full 8-bit

resolution RGB color separating capability are utilized for the pre-processing, the inference accuracy can be over 90% even only after 60 epochs (Supplementary Fig. 38).

## Extension to reservoir computing system

In addition to the application in the human visual system, the wavelength-dependent photo-synaptic property can also be leveraged in the design of a reservoir computing (RC) system. RC is based on a recurrent neural network that enables dynamic processing of continuous or temporal data. A typical RC system consists of input, reservoir, and readout layers (Fig. 8a). The reservoir layer is responsible for implementing fading memory and nonlinear dynamics, which allow inputs to be mapped onto a high-dimensional space represented by reservoir states. These functions reduce the need for a large number of input nodes when processing substantial amounts of data, making RC suitable for handling time-dependent data, including natural language processing, and weather forecasting, and financial forecasting[68-70].

The reservoir layer in the RC system requires the ability to reflect the present input as well as the preceding and recent inputs, so that the fading memory effect is essential. Therefore, the synaptic device should possess short-term memory characteristics as prerequisites for its implementation[71,72]. As shown in Fig. 8c, the CH-P-integrated memristor exhibits threshold switching behavior when subjected to a low voltage bias (e.g., <0.1 V), indicating its short-term memory characteristics without any non-volatile conductance change at this voltage condition.

As for the optimal pulse conditions, a certain amount of time must elapse without any stimulation after the pulse stream, allowing the device state to return to its original state. Moreover, an accumulation effect should gradually occur as multiple pulses are applied at short intervals. In Fig. 8d and Supplementary Fig. 39, the variation of PSC in the CH-P-integrated memristor is depicted in response to changes in the height (1–20 mW cm⁻²) and width (50 and 100 ms) of a single R

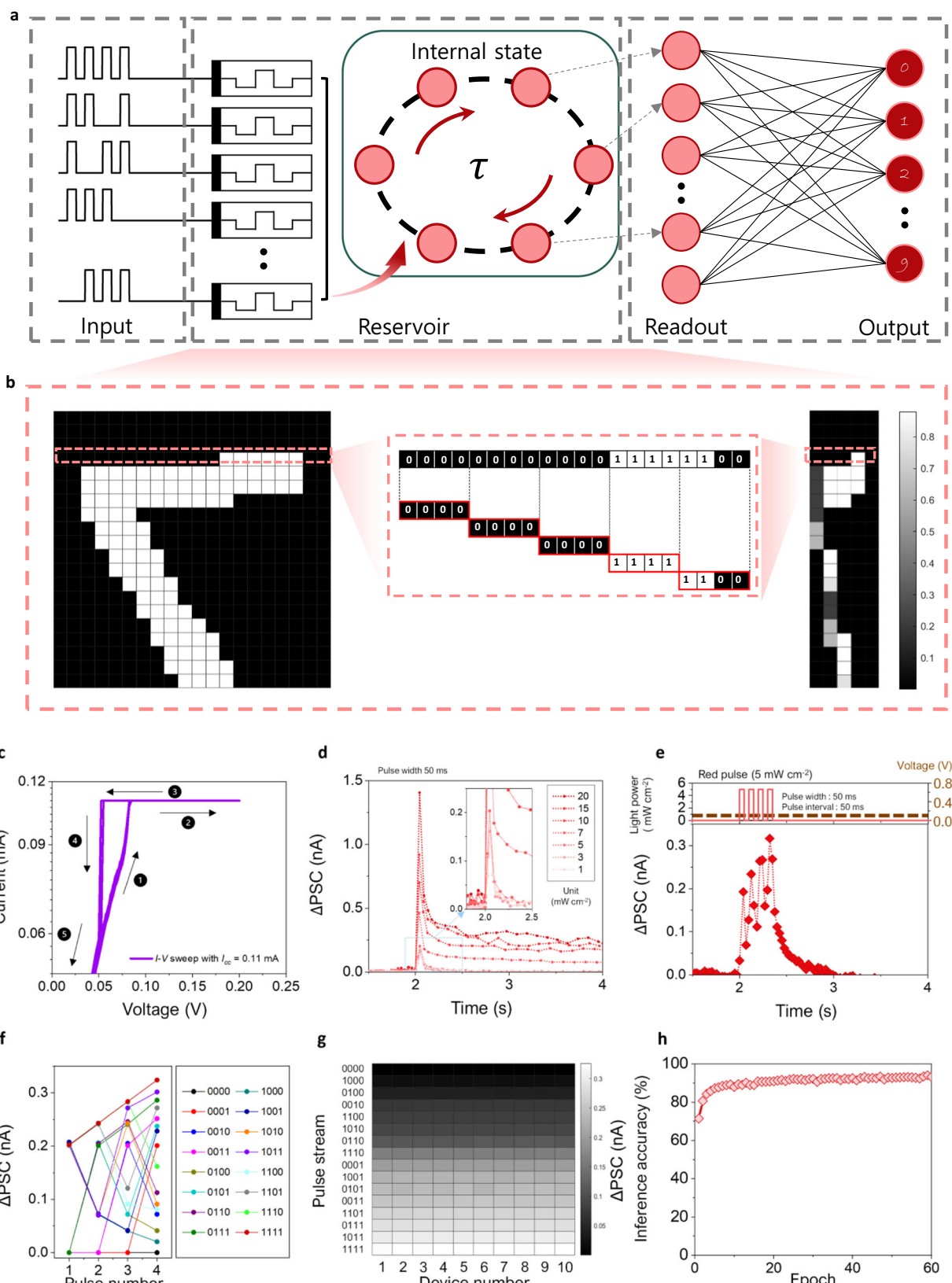

light pulse. It is observed that when using a R pulse with a width of 50 ms and an intensity below 5 mW cm$^{-2}$, the PSC decays to 0 without any conductance change within 0.3 s (Fig. 8d). Even when four consecutive pulses are applied (width: 50 ms, interval: 50 ms, and intensity: 5 mW cm$^{-2}$), the device exhibits a gradual increase in PSC, but no non-volatile conductance change is observed (Fig. 8e). This indicates

that these light pulse condition can be effectively employed to extend the applicability of CH-P-integrated memristors to the RC system.

Figure 8b illustrates that a modified National Institute of Standards and Technology (MNIST) pattern consisting of a total of 400 pixels (20 × 20) is compressed into a 100-pixel pattern (5 × 20) through the RC layer. As shown in Fig. 8f, g, the output current values of 16

**Fig. 8 | Optoelectronic synapses for light-programmable reservoir computing system. a** Schematic of the process flow in reservoir computing system, composed of input, reservoir, readout, and output layer. The number of pixels is minimized by encoding. **b** Original MNIST pattern consisting of 400 pixels (20 × 20) is compressed into a 100-pixel pattern (5 × 20) through RC layer. **c** Measured 20-cycle d.c. *I–V* characteristics of CH-P-integrated memristor at low voltages ($I_{cc}$ was set to 0.11 mA) exhibiting threshold switching. **d** PSC variation of CH-P-integrated memristor at a single R-light (450 nm) pulse depending on the amplitude of pulse (1–20 mW cm$^{-2}$). Inset figure is large magnification. **e** PSC variation of CH-P-integrated memristor at four consecutive R-light pulses. Amplitude and interval of R-light pulses are 5 mW cm$^{-2}$ and 50 ms. Pulse width in (**d**) and (**e**) is fixed at 50 ms. **f, g** PSC responses of CH-P-integrated memristor to 16 different cases of pulse stream. PSC data measured from 10 devices are presented in (**g**). Read voltage in (**d–g**) is 0.05 V. **h** Inference accuracy of the neural network with the light-programmable RC system.

possible cases (0000, 0001, 0010, ..., etc.), generated by four consecutive light pulses, are distinct from one another, and these temporally generated output current values effectively compress a significant amount of input information based on the time series. The data in Fig. 8g are from 10 devices, demonstrating the consistency and uniformity across the devices. The average and standard deviation of output current after each pulse stream condition are summarized in Supplementary Table 4.

The pattern recognition task in the network employing the RC system was conducted in two stages—RC and recognition systems. For reference, the MNIST pattern images were binarized using a thresholding technique and then were inputted into the reservoir framework. Four sequential pixels in a row of a MNIST pattern were grouped together into a single pixel value (Fig. 8b). This encoding process reduces the complexity of artificial neural network (ANN), as the output values from the reservoir layer are used for the input values of learning and inference tasks of ANN. The mapped pixels were applied to the ANN using NeuroSim framework. As synapses in ANN, CH-P-integrated memristors were also utilized, and B-light pulses were applied to update their weight values. The LTP/LTD characteristics of the CH-P-integrated memristor, such as nonlinearity and symmetricity under B-light pulses, are sufficient enough to achieve a high recognition rate, as shown in Fig. 8h. The implementation of the reservoir layer enables processing with a reduced input size. The single-layer perceptron, consisting of 100 input neurons and 10 output neurons even without a hidden layer, achieves impressive recognition rate of 94.1% after 60 epochs (Fig. 8h).

## Discussion
The color-discrimination characteristic is a crucial factor for the photonic synapse to have for achieving an artificial visual system; however, its demonstration has been limited to the three-terminal architecture devices, and the specific materials are required to exhibit the photosynaptic functionalities, consequently restraining its extensive application to various systems. We demonstrate a mechanism that can introduce the color-discriminating photo-synaptic functionality into a simple two-terminal memristor without any optical filters and complicated circuitry even regardless of the switching medium and electrode, which can extend its applicability. This feature is also beneficial to reducing the size and complexity of each pixel favorable to achieving an ultra-high-resolution system. This approach is possible with the asymmetric organic semiconductor, which has long-lasting strong molecular polarity in the photo-excited state. It can generate light-enhanced induced dipole moment within its interlayer, enhancing the electric field within the device at the light irradiation and thus making the light as an input signal of the memristor instead of the voltage. As a design principle, we prove that the photo-sensitive synaptic characteristics of the devices are maximized by increasing the exited state lifetime of the molecules and also adjustable by controlling the strength of their intrinsic dipole moment. Furthermore, their wavelength-dependence, originated from the spectral absorbance and excited state lifetime variation of the designed molecule, can be a basis of the color selectivity of the memristor. The prepared optoelectronic memristor-arrays show great potential as a component of the neuromorphic artificial visual system that can detect and pre-process the

colorful input signals like the retina, and also be an element of an intelligent light-programmable CNN with superior inference accuracy to the color images. In addition, the wavelength-dependent photosynaptic property can also be leveraged in the design of a light-programmable RC system. This approach is expected to be a platform technology, which can introduce the photo-synaptic functionality to various material-based artificial visual systems along with RC systems, with its wide applicability.

## Methods
### General synthetic procedure
*4-(di(naphthalen-2-yl)amino)-2-(1,4,5-triphenyl-1H-imidazol-2-yl)phenol* (DNH): Place a mixer of 4-bromo-2-(1,4,5-triphenyl-1*H*-imidazol-2-yl) phenol (1a in Supplementary Fig. 1a) (1.5 g, 3.21 mmol), di(naphthalen-2-yl)amine (2a in Supplementary Fig. 1a) (1.04 g, 3.98 mmol), Pd(OAc)$_2$ (0.02 g, 0.0996 mmol), P(*t*-Bu)$_3$ (0.39 mL, 0.39 mmol), and *t*-BuONa (1.02 g, 10.60 mmol) in a 100 mL two necked flask under an nitrogen atmosphere. Then 40 mL of dry toluene was added. After stirring for 15 h at 110 °C, the reaction mixer was cooled to room temperature. The reaction was quenched by water and extracted with ethyl acetate. The organic layer was collected and concentrated. The residue was purified via column chromatography by using hexane/dichloromethane (80:20 v/v) as eluent to afford yellow solid (1.85 g, 88%).

*N-(naphthalen-2-yl)-N-(3-(1,4,5-triphenyl-1H-imidazol-2-yl)phenyl) naphthalen-2-amine* (DN): Place a mixer of 2-(3-bromophenyl)-1,4,5-triphenyl-1*H*-imidazole (1b in Supplementary Fig. 1b) (1.50 g, 3.32 mmol), di(naphthalen-2-yl)amine (2a in Supplementary Fig. 1b) (1.07 g, 3.98 mmol), Pd(OAc)$_2$ (0.02 g, 0.0997 mmol), P(*t*-Bu)$_3$ (0.40 mL, 0.40 mmol), and *t*-BuONa (1.05 g, 10.96 mmol) in a 100 mL two necked flask under an nitrogen atmosphere. Then 40 mL of dry toluene was added. After stirring for 15 h at 110 °C, the reaction mixer was cooled to room temperature. The reaction was quenched by water and extracted with ethyl acetate. The organic layer was collected and concentrated. The residue was purified via column chromatography by using hexane/dichloromethane (80:20 v/v) as eluent to afford white solid (1.74 g, 82%).

*2-(4,5-bis(4-fluorophenyl)-1-phenyl-1H-imidazol-2-yl)-4-(di(naphtha-len-2-yl)amino)phenol* (DNH-F): Place a mixer of 2-(4,5-bis(4-fluor-ophenyl)-1-phenyl-1*H*-imidazol-2-yl)-4-bromophenol (1c in Supplementary Fig. 1c) (1.5 g, 2.98 mmol), di(naphthalen-2-yl)amine (2a in Supplementary Fig. 1c) (0.96 g, 3.58 mmol), Pd(OAc)$_2$ (0.02 g, 0.0894 mmol), P(*t*-Bu)$_3$ (0.36 mL, 0.36 mmol), and *t*-BuONa (0.95 g, 9.83 mmol) in a 100 mL two necked flask under an nitrogen atmosphere. Then 40 mL of dry toluene was added. After stirring for 15 h at 110 °C, the reaction mixer was cooled to room temperature. The reaction was quenched by water and extracted with ethyl acetate. The organic layer was collected and concentrated. The residue was purified via column chromatography by using hexane/dichloromethane (80:20 v/v) as eluent to afford yellow solid (1.80 g, 87%).

*N-(3-(4,5-bis(4-fluorophenyl)-1-phenyl-1H-imidazol-2-yl)phenyl)-N-(naphthalen-2-yl)naphthalen-2-amine* (DN-F): Place a mixer of 2-(3-bromophenyl)-4,5-bis(4-fluorophenyl)-1-phenyl-1*H*-imidazole (1d in Supplementary Fig. 1d) (1.5 g, 3.08 mmol), di(naphthalen-2-yl)amine (2a in Supplementary Fig. 1d) (1.0 g, 3.69 mmol), Pd(OAc)$_2$ (0.02 g, 0.0923 mmol), P(*t*-Bu)$_3$ (0.37 mL, 0.37 mmol), and *t*-BuONa (1.05 g,

10.56 mmol) in a 100 mL two necked flask under an nitrogen atmosphere. Then 40 mL of dry toluene was added. After stirring for 15 h at 110 °C, the reaction mixer was cooled to room temperature. The reaction was quenched by water and extracted with ethyl acetate. The organic layer was collected and concentrated. The residue was purified via column chromatography by using hexane/dichloromethane (80:20 v/v) as eluent to afford white solid (1.68 g, 81%).

*(E)-3-(4-(dimethylamino)phenyl)-1-(1-hydroxynaphthalen-2-yl)prop-2-en-1-one* (CH-M): 1.50 g (8.06 mmol) of 2-Acetyl-1-naphthol and 1.20 g of 4-(dimethylamino)benzaldehyde 8.06 mmol) were placed in 100 mL round-bottom flask and dissolved in 30 mL of ethanol. 0.662 mL of pyrrolidine was added to the solution and stirred overnight at room temperature. The color of solution turns red from yellow after the addition of pyrrolidine. After the completion of the reaction, the mixture was poured into water, filtered, and washed with cooled ethanol. The filtrate was dried using $MgSO_4$, and the crude product was column-purified by silica-gel column chromatography using EA and n-hexane as eluent (25:75 v/v). 1.93 g of product was obtained after the purification (Y = 64.5%).

*(E)-3-(4-(dimethylamino)phenyl)-1-(naphthalen-2-yl)prop-2-en-1-one* (C-M): 2-Acetylnapthalene (0.50 g, 2.93 mmol) and 4-dimethylamino benzaldehyde (0.44 g, 2.93 mmol) were placed in 100 mL round-bottom flask and charged with 10 mL of ethanol. 0.21 g (2.93 mmol) of pyrrolidine was added to the solution and stirred overnight at room temperature. After completion of the reaction, the solvent was removed under reduced pressure, the residue was dissolved in ethyl acetate and washed with water. The organic layer was separated and dried over $Na_2SO_4$. The organic layer was evaporated in vacuum and the crude product was purified by column chromatography (Hexane/EtOAc = 10:1) to give C-M (600 mg) as yellow solid in 66% yield.

*(E)-3-(4-(diphenylamino)phenyl)-1-(1-hydroxynaphthalen-2-yl)prop-2-en-1-one* (CH-P): 1.50 g (8.06 mmol) of 2-Acetyl-1-naphthol and 2.20 g of 4-(diphenylamino)benzaldehyde 8.06 mmol) were placed in 100 mL round-bottom flask and dissolved in 30 mL of ethanol. 0.662 mL of pyrrodine was added to the solution and stirred overnight at room temperature. The color of solution turns red from yellow after the addition of pyrrolidine. After the completion of the reaction, the mixture was poured into water and filtered, washed with cooled ethanol. The filtrate was dried over $MgSO_4$, and the crude product was column-purified by silica-gel column chromatography using EA and n-hexane as eluent (20:80 v/v). 1.34 g of product was obtained after the purification (Y = 37.7%).

*(E)-3-(4-(diphenylamino)phenyl)-1-(naphthalen-2-yl)prop-2-en-1-one* (C-P): 2-Acetylnapthalene (0.50 g, 2.93 mmol) and 4-diphenylamino benzaldehyde (0.80 g, 2.93 mmol) were placed in 100 mL round-bottom flask and charged with 10 mL of ethanol. 0.21 g (2.93 mmol) of pyrrolidine was added to the solution and stirred overnight at room temperature. After completion of the reaction, the solvent was removed under reduced pressure, the residue was dissolved in ethyl acetate and washed with water. The organic layer was separated and dried over $Na_2SO_4$. The organic layer was evaporated in vacuum and the crude product was purified by column chromatography (Hexane/EtOAc = 10:1) to give C-P (720 mg) as yellow solid in 58% yield. $^1H$ NMR (600 MHz, $CDCl_3$) δ 8.54 (s, 1H), 8.11 (dd, J = 8.4, 1.8 Hz, 1H), 7.99 (d, J = 8.4 Hz, 1H), 7.94 (d, J = 9 Hz, 1H), 7.90 (d, J = 8.4 Hz, 1H), 7.85 (d, J = 15.6 Hz, 1H), 7.62–7.58 (m, 2H), 7.56–7.54 (m, 3H), 7.33–7.31 (m, 4H), 7.18–7.17 (m, 4H), 7.14–7.11 (m, 2H), 7.07 (d, J = 8.4 Hz, 2H). $^{13}C\{^1H\}$ NMR (150 MHz, $CDCl_3$) δ 190.40, 150.31, 146.95, 144.75, 136.09, 135.47, 132.71, 129.91, 129.77, 129.64, 129.60, 128.58, 128.32, 128.02, 127.92, 126.81, 125.60, 124.70, 124.26, 121.70, 119.56. HRMS (ESI) *m/z*: calculated for $C_{31}H_{23}NO$ $[M]^+$ 425.1780, Found: 425.1771.

## $NiO_x$ NP synthesis

Synthesis of $NiO_x$ NPs was performed by adopting the chemical precipitation method from our previous studies[51]. 99.999% nickel(II) nitrate hexahydrate ($Ni(NO_3)_2 \cdot 6H_2O$) was purchased from Sigma-Aldrich, polyethylene glycol (PEG) and sodium hydroxide (NaOH) were purchased from Alfa Aesar and Merck. All solutions were prepared by dissolving them into deionized water (DI) and the entire reaction process was performed at room temperature. $Ni(NO_3)_26H_2O$ was dispersed in DI water to obtain a dark green solution, and PEG was added during continuous magnetic stirring. Then, NaOH was slowly added dropwise at a rate of about 10 μl per drop. This process was carried out under vigorous stirring until the pH of the solution was adjusted to 10. After the completion of the reaction, the colloidal precipitate was washed with DI water several times to remove residual impurities and chemicals. The colloidal precipitate was then baked at 80 °C for overnight and calcined in a furnace at 285 °C for 2 h to obtain black powder. The produced black powder was pulverized into finer particles through a grinder.

## Crossbar array fabrication

FTO on glass substrate was used as bottom transparent electrode. To prepare a crossbar array, FTO-coated glass substrate underwent cleaning with acetone and IPA before being spin-coated with the solution of photoresist (PR) AZ5214 at a speed of 1000 rpm for 1 min. The deposited PR film was then subjected to a soft bake at 100 °C for 5 min. A patterned mask with 8 lines, of which line widths were 100 μm, was placed on top of the PR film, which was then exposed to UV-light for 7 s. After exposure, the PR film was developed for 1 min in AZ 300 MIF solution, rinsed with DI water, and subjected to a hard bake at 120 °C for 5 min. The patterned positive PR-coated FTO film was etched for 20 min using $CH_4$ gas in a reactive ion etcher (with a gas flow rate of 90 sccm and reflective power of 300 W). The residual PR was removed using acetone and IPA. Next, the NiO thin film layer was deposited onto the patterned FTO substrate using spin-coating at 2000 rpm for 30 s, followed by annealing at 550 °C for 2 h in a furnace with ambient air. Organic molecules (DNH, DN, DNH-F, DN-F, CH-M, C-M, CH-P, and C-P) were dissolved in chlorobenzene and coated on the NiO layer (2000 rpm for 30 s) and annealed on the hotplate at 100 °C for 10 min. Finally, Ag top electrode was deposited using a shadow mask with 8 lines, of which line widths were 100 μm, by thermal evaporator in a vacuum environment ($<1 \times 10^{-6}$ Torr).

## Device measurement

All the measurements were conducted inside a dark shield box under ambient air condition. For the characterization of resistive switching and synaptic behaviors, d.c. *I–V* characteristics, EPSC, PPF, and LTP/LTD were measured using a probe station equipped with a Keithley 2636B semiconductor parameter analyzer. *I–V* characteristic measurements were conducted using two methods: double sweep mode (a unidirectional cyclic loop *I–V* measurement system, possible in either positive or negative direction) and cycle sweep mode (a bidirectional cyclic loop *I–V* measurement system). In addition, an appropriate compliance current (15 mA was set for *I–V* cycle sweep measurement) was set to prevent device breakdown and the formation of permanent filaments. Synaptic behavior measurements were performed using the SP-300 model potentiostat (BioLogic) to generate optical pulses, and the behavior induced by the pulses was monitored in real-time by applying a constant read bias of 0.05 V in the time domain. Optical measurements were carried out by irradiating the optical beam onto the bottom electrode, FTO, of the device. The light sources for the optical pulses were light emitting diodes (LEDs), with available wavelengths of 365 nm (UV), 450 nm (blue), 525 nm (green), and 630 nm (red). The power of the light was calibrated using a photodiode sensor, the PD300 (Ophir), with ranges of 1 to 3 mW cm$^{-2}$ for UV and 1 to 20 mW cm$^{-2}$ for RGB. To characterize a single device, light is exposed through an aperture (1 mm diameter). For the RGB letter pattern, optical masks are used. The memristor pixels in a letter with the same color were illuminated simultaneously, and their output currents were

sequentially read. Retention measurements were conducted on a hotplate, in ambient air environment, and within a dark shield box where light was blocked. The retention behaviors at both HRS and LRS were measured. Retention at LRS was taken into two conditions: after 50 B-light pulses and UV-light pulses. Cycle endurance was conducted in 50-cycle segments. This was to prevent communication buffer issues between the computer and the equipment (Keithley 2636B) during long-term measurements. The transfer curve ($V_{GS}$–$I_D$) of FET was measured in two conditions: under light irradiation and dark condition. The dark condition was first conducted, and then light irradiation was performed. The light exposure lasted from the beginning to the end of the measurement and was maintained at a constant intensity.

**Neural network simulation**

The feed-forward process of CNN has two parts of operations: feature extraction and classification. For the feature extraction, pre-processed color images, of which each pixel is separated into three RGB color channels, are convoluted with kernels, generating the feature maps, and their sizes are reduced through the pooling layer. To describe the cases having the overlapped RGB components between the colors after the pre-process, the dataset of the three-color channels in the CIFAR-10 is modified using Python language. The CIFAR-10 color input images and convolutional kernels are composed of $32 \times 32$ and $3 \times 3$ pixels. Each pixel of the input image and the kernel is represented as a voltage ($V$) and a synaptic weight ($W$). The synaptic weight is represented by the conductance difference of two equivalent memristors to demonstrate both positive and negative weight values, and the weights are updated by the unidirectional or bidirectional method as explained in Supplementary Note 11. The results of the convolution ($I = W \times V$) are represented as the current ($I$) in the feature map and converted to the voltage by the ReLU activation function. Pooling reduces the size of the map with a highlight. This cycle is repeated three times. The dimensions of the kernels for 1st, 2nd, and 3rd convolution layers are 3 (the number of channels) $\times$ 3 (kernel width) $\times$ 3 (kernel height) $\times$128 (the number of kernels), $128 \times 3 \times 3 \times 256$, and $256 \times 3 \times 3 \times 512$. The number of feature maps in 1st, 2nd, and 3rd convolution layers are 128, 256, and 512. After 1st, 2nd, and 3rd pooling, the dimensions of the feature maps are reduced from $32 \times 32$, $16 \times 16$, and $8 \times 8$ to $16 \times 16$, $8 \times 8$, and $4 \times 4$. After the final 3rd pooling layer, the voltage signals are flattened to a $1 \times 8192$ array and transferred to the input layer of the FCLs for the classification. The FCLs are composed of 8192 input neurons, 1024 hidden neurons, and 10 output neurons. In software-based artificial neural network (ANN), both positive and negative values are utilized for the weight values of individual cells. However, in hardware crossbar array ANN, the conductance of the memristor ($G$), which represents the synaptic weight ($W$), cannot have negative values. Therefore, to demonstrate both positive and negative weight values in hardware ANNs, the synaptic weight of each cell is represented by the conductance difference ($W = G^+ - G^-$) of a pair of two equivalent synapses, of which the conductance values are represented by $G^+$ and $G^-$ ($G^+ > 0$ and $G^- > 0$). The input voltage signals generate the current values by the weighted-sum operation ($I = \sum W \times V$) with the synaptic weights between input and hidden neurons. The current signals are converted into the voltage signals by the ReLU activation function and added to the hidden layer as input signals. Finally, these voltage signals generate the current at the output layer by the weighted-sum operation with the synaptic weights between hidden and output neurons. The backpropagation operation is performed by comparing output layer value ($V_o$), obtained by converting the current at the output layer to the voltage using the ReLU activation function, and label value ($K$). Synaptic weights are updated from output layer to the first convolution layer by minimizing the error between output and label values ($\delta = K - V_o$).

## Data availability

All data that support the findings of this study are provided in the paper and the Supplementary Information. Further information including the raw datasets used for the presented analysis within the current study is available from the corresponding authors upon request. Correspondence and requests for materials should be addressed to H.J.P.

## Code availability

DNN+ NeuroSim used for neural network simulation can be accessed through the GitHub public repository (https://github.com/neurosim/DNN_NeuroSim_V2.0) and ref. 67.

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

## Acknowledgements

The authors acknowledge funding from the National Research Foundation of Korea (2022M3I7A3050450 to H.J.P.; 2020R1A2C2010342 to H.J.P.; 2022R1A4A3032913 to H.J.P., and 2021R1A2C1008375 to S.P.).

## Author contributions

J.L. conducted all experimental and simulation works. B.H.J. conducted experimental works related to photonics. E.K. synthesized and analyzed molecules. D.K. analyzed data and designed reservoir computing. H.K. analyzed data. H.J.P. and S.P. conceived and directed the research project. H.J.P. supervised the project. H.J.P., J.L., and S.P. wrote the manuscript. All authors contributed to the preparation of the manuscript.

## Competing interests

The authors declare no competing interests.
