## [Peer Review File · Nature Communications]

Reviewers' comments:

Reviewer #1 (Remarks to the Author):

In this manuscript, the authors report a memristor device with light enhancement, which realizes the reception of different colors by using different semiconductor materials and is used for image recognition under multi-colors. However, there are many similar researches on memristors that can feel illumination, and there are even memristors that can be controlled by light, so the characteristics of this device are not put forward for the first time (Nano Lett. 2021, 21, 14, 6087–6094). The materials used in this work are not new as well. In addition, it is not the first time that the application of neural network with multiple colors to improve the accuracy of image recognition has been proposed. [Adv. Mater. 2022, 2108979; Nat Commun 2018, 9, 5106] Therefore, the innovation of this article is not sufficient enough and it is not suggested to be published in Nature Communications. Some other questions are listed below:

1. In the first part, "in this context ... which form a hierarchical biostructure." has no reference, and the authors should supplement the relevant literature.
2. In the explanation of the phenomenon of light enhancement, the authors claim that the device will produce potential difference V_b under short circuit. What causes this potential difference? What are the requirements? Is V_b a photogenerated electromotive force similar to a solar cell generated under light?
3. If the innovation of this manuscript is to propose a series of materials, then the framework of the article needs to be revised to highlight the importance of the material system.
4. The application of this paper only shows that this synapse is used to construct the input layer for color recognition, while other network frameworks are existed, and it does not show the difference between this device and other synapses in other network structures except the input layer. It is difficult to reflect the uniqueness of this device in neural network.
5. More synaptic characteristics under photoelectric control and the proposal of a new architecture network based on these characteristic should be provided.
6. Regarding the light enhancement part of the device, the mechanism is not solid and well discussed. For instance, simulation of the underlining mechanism is suggested to be supplemented.

Reviewer #2 (Remarks to the Author):

<Overall comment>

The author has developed two-terminal optical memristors utilizing various combinations of metals and organic materials. Through experimentation, the author has claimed that they have demonstrated two memristors with distinct functionalities - a multispectral optical memristor for sensing and an optically programmable memristor for computing in a crossbar configuration. The artificial vision system RGB image encoding for pre-processing, and hardware implementation of matrix multiplication programmable by optical flux attract great interests, while it is hardly convinced that this work holds promise for use in multispectral in-sensor vision computing due to multiple limitations. The manuscript has a few shortcomings that need to be addressed. The manuscript contains redundant information with too complex sentences, and several portions of the methodology are not clear.

<Questions>

1. Authors claim that their device is “multispectral color-cognitive memristor” which is the main theme of this manuscript. It is clear that their molecule provides spectral resolution between UV and visible. However, it is hard to believe the color-cognition function in visible range since there is no spectral selectivity, but is only shows non-uniform broad absorption spectrum. Thus, this device would only work for color cognition under very specific conditions such as illumination of identical intensity RGB light with specific wavelength to the device simultaneously etc. For example, CH-M based device cannot distinguish between ~400nm and ~500nm wavelength light with identical intensity where the absorption coefficient is the same. And, it cannot distinguish 525nm wavelength light from 450nm wavelength light with half intensity of it since the absorption coefficient is half at 525nm. In my opinion, this is just memristor that can be programed by visible light unless individual RGB sensors with color filters are integrated as pixel.

2. Some two-terminal optical memristors with multi-spectral (in visible spectra) responsivity have already been reported in other literature¹⁻³. What are the advantages of the proposed device over these references? Please consider these references in Supplementary Table. 1 as well as the ON/OFF ratio, which is a key metric to realize the effective memristor-based multiplication⁴.

1. Pei, Yifei, et al. "Artificial visual perception nervous system based on low-dimensional material photoelectric memristors." ACS nano 15.11 (2021): 17319-17326.

2. Wang, Wenxiao, et al. "Artificial optoelectronic synapses based on $\text{TiN}_x\text{O}_{2-x}/\text{MoS}_2$ heterojunction for neuromorphic computing and visual system." *Advanced Functional Materials* 31.34 (2021): 2101201.

3. Hu, Lingxiang, et al. "All-optically controlled memristor for optoelectronic neuromorphic computing." *Advanced Functional Materials* 31.4 (2021): 2005582.

4. Chen, Pai-Yu, Xiaochen Peng, and Shimeng Yu. "NeuroSim: A circuit-level macro model for benchmarking neuro-inspired architectures in online learning." *IEEE Transactions on Computer-Aided Design of Integrated Circuits and Systems* 37.12 (2018): 3067-3080.

3. Figure 3j shows the retention time of CH-M at LRS only, thus the author needs to include more information above to claim the simulated memristor crossbar based on the DNH-F memristors. What are the retention times of both CH-M and DNH-F memristors at HRS and LRS, respectively?

4. The author has provided various device structures with different material combinations, some of which may be redundant in guiding readers to the conclusion. Since the artificial vision system includes only CH-M and DNH-F memristors, other device structures and material combinations can be regarded as optimization/improvement procedures for the ultimate artificial vision system. It is recommended that the main figures be rearranged, and the corresponding texts revised accordingly, considering the ultimate application and the potential universal applicability of the device.

5. The advantages of optical programming mentioned on page 17 may not be applicable to this work, as the corresponding ref. 19 involves a three-terminal device with a 2D material. The author should consider providing other advantages of optical programming over electrical programming of memristors (DNH-F) used in the crossbar structure. It is also challenging to focus the optical beam in a scaled-down memristor crossbar in nanometers to selectively program each memristor pixel, and electrical programming allows for nanosecond programming capability with compact electrical circuitry to minimize parasitic capacitance and inductance.

6. Author explained that every color input image is separated into RGB three channels through the CH-M-integrated memristor array as a pre-processing. However, it is almost impossible to perform this task using the demonstrated device since the image color is super-positioned spectrum that cannot be resolved using the absorption spectrum of CH-M based memristor. For example, if Red and Green light is illuminated together, then how this single device can recognize the color without color filter? If there is any mechanism to achieve this using the proposed device, it should be clearly explained.

7. Authors claimed that the synaptic weight of each cell is defined by the conductance difference between two equivalent DNH-F devices to represent negative weights. Did author employe negative weight value for the neural network simulation? If so, how can you generate the negative weight value just using two memristors? Based on author's schematic description in figure 7d, it is not possible to create negative weight values at each pixels, but it shows summation at the column and process it through differential amplifier. This should be clearly explained.

8. What are Gmax and Gmin used in the NeuroSim simulation? Is the base experimental LTP/LTD (Supplementary Fig. 26a) the same result with Fig. 5h? The ON/OFF ratio of Fig. 5h is insufficient to realize the memristor-based matrix multiplication as mentioned in the NeuroSim paper1.

1. Chen, Pai-Yu, Xiaochen Peng, and Shimeng Yu. "NeuroSim: A circuit-level macro model for benchmarking neuro-inspired architectures in online learning." IEEE Transactions on Computer-Aided Design of Integrated Circuits and Systems 37.12 (2018): 3067-3080.

9. In method section, there is no description about characterization. Organic device is usually unstable under ambient condition. Is this device measured under ambient condition or under nitrogen condition? If it is measured in the glove box, how does it works under the ambient conditions?

Below are the minor comments.

1. Please clarify and the terminologies, EPSC and PSC that were randomly used in the manuscript.

2. There are too many run-on and long sentences that are hard to be understood. Please split each information and simplify the sentences to improve the readability.

3. There is typo in page 9 "OEFT"

Reviewer #3 (Remarks to the Author):

In this manuscript, the authors reported that the introduction of organic semiconductor layer in an NiO-based semi-transparent memristor, which has a long lasting strong molecular polarity in the photo-excited state. Some comments are as follows-

1. The authors reported that the bipolar resistive switching (BRS) in their NiO based-memristor is possible because of migration of "Ag-ions" (active electrode), whereas the device with the inert electrode "Au" does not exhibit BRS. But if we compare the compliance currents (CC) in Fig. 3e and supplementary Fig. 22, then we observed a large difference (1 million times) between them. Is that possible if we apply the same amount (higher) of CC then BRS may be occurred in Au-based device?
2. In supplementary table-1, the authors compared the non-linearity (NL) and symmetricity of the devices reported in literature, which have been estimated by extracting LTP/LTD data. I suggest that NOT to put those references in which NLs of both LTP and LTD curves did not mention (such as ref. 55, 56, 60, 65 etc.). Also, I am just curious and want to confirm that the symmetricity equation (supplementary note 7) will be precisely applicable in those cases (supplementary ref. 13, 52) where the number of applied pulses are different for LTP/LTD or not?
3. In the section of 'Emulation of human visual system', the authors reported that the patterning of RGB- colored letters is illustrated by preparing a photonic synaptic array of 8x8 pixels. Is this an image array or memristive device array? In 'device fabrication' section, the device array preparation has not mentioned. Also, I will suggest the authors to specify the annealing environment for NiO layer.
4. The authors must provide the JCPDS file number (as a reference) to match the intensity and position of diffraction peaks in XRD pattern.
5. If the fabricated memristor is not forming-free, then the authors must provide the required forming (the very first set) process in Fig. 3e.
6. The authors must mention the calculated values of relaxation times of PPF-index curve during the fast and slow phase of exponential decay.
7. The authors must mention the size of 'laser spot area on device', which was fixed during the optical measurements.
8. There are some spelling mistakes in p.11.

Followings are detailed responses to reviewer's comments:

1. About reviewer #1's comments:

- We appreciate the reviewer #1's feedback, as it has provided us with an opportunity to re-evaluate our design principles and review the relevant literature again. The reviewer expressed concerns that *the innovation of this article is not sufficient enough and it is not suggested to be published* (as written in reviewer's comments). We would like to briefly address the reviewer's concerns as follows:

- 1) Firstly, the reviewer noted that there are similar research studies already published. However, we would like to emphasize that our photonic memristor synapse exhibits **RGB color recognition ability**, which **has not been demonstrated by any other 2-terminal photonic memristor** to date, as we explain in the point-to-point responses. Our approach is also **universal**, regardless of the material or medium used, which **has never been demonstrated before**.
- 2) Secondly, the reviewer mentioned that the materials used in our work are not new. However, **the main materials**, used to demonstrate photonic synapse are **CH-P** for visible and **DNH-F** for UV, **are entirely new** and have not been reported previously, **as confirmed by SciFinder evaluation** in the following detailed responses. We would also like to highlight that **CH-P is newly designed for this revision** (Fig. 1, page 26-27, Supplementary Fig. 2, 4, 17, 18, 19, 20, 22, 25, 26, and Supplementary Note 7-8).
- 3) Thirdly, the reviewer commented that other researchers have proposed neural networks with multiple colors. However, those are based on 3-terminal-transistor, and we developed a neural network based on **2-terminal memristors having a clear advantage in high integration density compared to the 3-terminal system**. Moreover, **we designed a new neural network, light-programmable reservoir computing system** (Fig. 8, page 21-23, and Supplementary Fig. 37), **for this revision**.
- 4) Fourth, **for this revision, we conducted a new Mott-Schottky analysis** (Fig. 1o and page 7, 8, 18), which further **strengthens the understanding of the operational mechanism** of our approach.

Followings are in-depth explanations for your comments.

- **Review summary 1/3:** *In this manuscript, the authors report a memristor device with light enhancement, which realizes the reception of different colors by using different semiconductor materials and is used for image recognition under multi-colors. However, there are many similar researches on memristors that can feel illumination, and there are even memristors that can be controlled by light, so the characteristics of this device are not put forward for the first time (Nano Lett. 2021, 21, 14, 6087–6094).*

Our response: The **characteristics** of our device, **demonstrated for the first time in this work**, **are not the properties that can feel illumination and that can be controlled by light**, **as reviewer 1 commented**, but **the color-recognition ability in a 2-terminal configuration**, along with **the universality of our approach for various mediums**. The reference, “*Nano Lett.* 2021, 21, 14, 6087–6094”, quoted as an example, is also already included in Supplementary Table 1 for literature survey, and it does not possess color-recognition ability. To elaborate further:

- 1) **Our work presents the first demonstration of a 2-terminal photonic memristor synapse with RGB color recognition ability**. As summarized in the 2-terminal-device section of Supplementary Table 1, 2-terminal photonic memristor synapses, reported in the literature, lacked RGB color-discrimination ability. While several research works on 3-terminal synaptic devices capable of recognizing RGB colors have been reported, our work firstly demonstrates the color-recognizable 2-terminal memristor with the advantage of higher integration density compared to 3-terminal devices.
- 2) **Our approach exhibits universality irrespective of the material or medium**. As depicted in Figure 5e,f, our mechanism does not depend on the properties of the medium or electrode, making it applicable to diverse material systems. In contrast, **most optoelectronic synapses reported in the literature, both 2-terminal and 3-terminal structures, require specific material combinations for their medium or electrodes** to exhibit photo-synaptic functionalities, limiting their applications to various systems.

- **Review summary 2/3:** *The materials used in this work are not new as well.*

Our response: The **main materials** used to demonstrate photonic synapse in our work are **CH-P** for visible light-responding photonic synapse and **DNH-F** for UV light-responding photonic synapse. These materials are **entirely new** and **have not been reported previously**, as **confirmed by SciFinder evaluation** in the following paragraphs. Notably, **CH-P has been newly designed for this revision**, with the aim of further enhancing the RGB-color recognition ability compared to CH-M, which was used to provide RGB-color recognition ability in the previous manuscript.

The materials in this work are classified into the following **4 types** (It is worth mentioning that DN, DN-F, C-M, and C-P, which do not contain "H" in their name, are used only for comparison purposes to highlight the fact that materials should be able to undergo the ESIPT process in order to exhibit the photo-functionality proposed in this work.).

	Type	Main material (ESIPT)	Just comparison (non-ESIPT)	
1)	UV-synapse	DNH	DN	
2)	UV-synapse / enhanced	DNH-F	DN-F	DNH-F is a new material
3)	Visible-synapse	CH-M	C-M	
4)	Visible-synapse / enhanced	CH-P	C-P	CH-P is a new material. CH-P is newly designed and synthesized for this revision.

1) DNH and DN:

We introduced the concept of utilizing the enhanced dipole moment effect of the organic semiconductor with extended lifetime upon excitation for photovoltaic cells using DNH and DN, as reported in *Adv. Funct. Mater.* 2021, 31, 2007180. Hence, DNH and DN (UV light-responding) are employed as starting materials here to propose the enhanced dipole moment effect at the light-exposure, which imparts our photo-synaptic properties. This is explicitly mentioned in the text (“*DNH and DN were designed in our previous work³¹, and DNH-F and DN-F are newly prepared to enhance the photo-responsivity of the device in the UV wavelength region.*” In page 4). The following SciFinder search refers to our previous article (*Adv. Funct. Mater.* 2021, 31, 2007180).

DNH: UV-synapse

DN: comparison of DNH

2) **DNH-F** and **DN-F**:

DNH-F is newly designed ESIPT material to enhance the intrinsic dipole moment of DNH, consequently enhancing the UV-responding synaptic property. DN-F is a just its comparison to show the necessity of ESIPT process in our operation mechanism. **The following SciFinder search confirm that they have not been reported previously.**

DNH-F: UV-synapse / enhanced

CAS SciFinder® Task History
 April 13, 2023, 11:23AM
 Initiating Search
 Substances: 1
 Filtered By:
 Structure Match: As Drawn
 Search Tasks
 Task Search Type View
 Exported: Returned Substance Results + Filters (0) Substances View Results

DN-F: comparison of DNH-F

CAS SciFinder® Task History
 April 13, 2023, 11:32AM
 Initiating Search
 Substances: 1
 Filtered By:
 Structure Match: As Drawn
 Search Tasks
 Task Search Type View
 Exported: Returned Substance Results + Filters (0) Substances View Results

3) **CH-M** and **C-M**:

CH-M is prepared to extend the UV-responsibility of the photonic synapse to visible range to recognize RGB colors, and **CM** is its non-ESIPT comparison. Although we firstly utilized **CH-M** molecule to the synaptic memristor, we got to know that it has been reported in several for other purpose. Therefore, we newly designed and synthesized **CH-P** for this revision, which has an enhanced phot-responsivity.

CH-M: visible-synapse

CAS SciFinder® Task History
 April 13, 2023, 11:19AM
 Initiating Search
 Substances: 1
 Filtered By:
 Structure Match: As Drawn
 Search Tasks
 Task Search Type View
 Returned Substance Results + Filters (4) Substances View Results

Task	Search Type	View
1	Substances	View Results
Synthesis of fluorescent core containing chalcone structure		
By: Zhu, Hailiang; Chen, Jian; Qi, Yalin; Zhang, Bo; Wang, Baoshong; China, CN111029827 A 2020-04-21 Language: Chinese, Database: Cqplus		
2	Substances	View Results
Synthesis and antimicrobial activity of 2-(substituted benzylidene)-6,7-benzocoumaran-3-one and its dibromide		
By: Kadu, V. B.; Doshi, A. G.; Oriental Journal of Chemistry (1997), 13(3), 281-284 Language: English, Database: Cqplus		
3	Substances	View Results
Synthesis and antimicrobial activity of 2-amino-4-(2-hydroxy-3,4-benzophenyl)-6-(substituted phenyl)pyrimidines and 2[1H]-pyrimidinones		
By: Kadu, V. B.; Doshi, A. G.; Research Journal of Chemistry and Environment (1998), 2(1), 69-71 Language: English, Database: Cqplus		
4	Substances	View Results
Microwave initiated synthesis of pyrimidine analogues and study of their antimicrobial properties		
By: Lak, Komand; Pathwal, L. J.; Bagade, M. B.; American Journal of PharmTech Research (2014), 4(3), 646-656 Language: English, Database: Cqplus		

C-M: comparison of CH-M

CAS SciFinder® Task History
 April 13, 2023, 11:47AM
 Initiating Search
 Substances: 1
 Filtered By:
 Structure Match: As Drawn
 Search Tasks
 Task Search Type View
 Returned Substance Results + Filters (2) Substances View Results

Task	Search Type	View
1	Substances	View Results
Supersensitization of photographic silver halide emulsions		
By: Jones, Joan E.; United States, US2860984 1958-11-18 Language: Undetermined, Database: Cqplus		
2	Substances	View Results
DFT analysis on spectral and NLO properties of (ZE)-3-(4-(dimethylamino) phenyl)-1-(naphthalen-2-yl) prop-2-en-1-one; a D-rt-A chalcone derivative and its docking studies as a potent hepatoprotective agent		
By: Lakshmi, C. S.; Nair, Balachandran, S.; Anil, Dhas D.; Ronalds, Anif A.; Hubert, Joe L.; Chemical Data Collections (2019), 20, 100205 Language: English, Database: Cqplus		

- 4) **CH-P** and C-P (Fig. 1, Supplementary Fig. 2, 4, 17, 18, 19, 20, 22, 25, 26, Supplementary Note 7-8): **CH-P is newly designed ESIPT material for this revision to enhance the photo-functionality in visible region.** This is the core material of our work for visible color recognition synapse. C-P is a just its comparison to show the necessity of ESIPT process. **The following SciFinder search confirms that CH-P has not been reported previously.**

- **Review summary 3/3:** *In addition, it is not the first time that the application of neural network with multiple colors to improve the accuracy of image recognition has been proposed. [Adv. Mater. 2022, 2108979; Nat Commun 2018, 9, 5106]*

Our response: The reported **hardware neural networks in the literature** so far, which have utilized multiple colors to enhance image recognition accuracy, including the articles [Adv. Mater. 2022, 2108979; Nat. Commun. 2018, 9, 5106], **are based on 3-terminal-transistor and inherently rely on specific materials,** limiting their integration into current technologies. As mentioned in the introduction, the 2-terminal-array offers a clear advantage in high integration density compared to the 3-terminal system. However, **an artificial neural network vision system based on a 2-terminal memristor array has not been successfully demonstrated until now.** Our work on color-recognizable memristors has led us to propose, **for the first time, an artificial neural network vision system based on a 2-terminal memristor array.** Furthermore, the freedom to select the switching medium and electrode enables easy integration of this work into various existing technologies.

Moreover, **for this revision,** we designed **a new neural network, light-programmable reservoir computing system** (Fig. 8, page 21-23, Supplementary Fig. 37). Please refer to answers to your comment 5.

- **Review comment 1:** *In the first part, "in this context ... which form a hierarchical biostructure." has no reference, and the authors should supplement the relevant literature.*

Our response: Thank you for your comment to make our manuscript more comprehensive. We added new references about a hierarchical biostructure of retina. Please refer to the reference [4].
[4] Fairchild, M. D. Color appearance models. (John Wiley & Sons, Ltd, 2013).

- **Review comment 2:** *In the explanation of the phenomenon of light enhancement, the authors claim that the device will produce potential difference V_b under short circuit. What causes this potential difference? What are the requirements? Is V_b a photogenerated electromotive force similar to a solar cell generated under light?*

Our response: As explained in the manuscript (page 6), the potential difference V_b is from the work function difference between top and bottom electrodes, and this work function difference between electrodes in metal-insulator-metal (MIM) memristor structure produces the built-in electric field (E_b) within the device at the equilibrium, regardless of light exposure. Therefore, V_b is not a photogenerated electromotive force similar to a solar cell generated under light.

Under this E_b , electron clouds within a polar molecule are oriented to the electric field, producing induced dipole moment, and the accumulative induced dipole moments of the molecules within the film can change the work-function of neighboring electrode (e.g. decreasing the work-function of Ag as depicted in Fig. 1m in this work), consequently intensifying the built-in potential ($V_b + V_d$) (right in Fig. 1n) and thus electric field within the device ($E_b + E_d$). Because the designed ESIPT molecules themselves (DNH, DNH-F, CH-M, and CH-P) have higher intrinsic dipole moment at the photo-excited state and this characteristic can be effective due to its extended lifetime at the excitation through ESIPT, the accumulative induced dipole moments within the film, can be further increased at the light exposure, consequently further changing the work-function of neighboring electrode. This light-generated additional work-function change provide additional increase of built-in potential V_{dl} (also additional increase of the electric field E_{dl}). V_{dl} and E_{dl} are higher than V_d and E_d , respectively. Therefore, electric field applied to the memristor device is increased from E_b+E_d to E_b+E_{dl} at the light signal, which means that light can be utilized as an input instead of the voltage. We revised main text in page 6 and Fig. 1m,n for better understanding, and furthermore newly performed Mott–Schottky analysis for this revision and confirm that the built-in potential increases with the light signal, as shown in Fig. 1o and page 7, 8, 18.

- **Review comment 3:** *If the innovation of this manuscript is to propose a series of materials, then the framework of the article needs to be revised to highlight the importance of the material system.*

Our response: We appreciate your comment to make our manuscript more comprehensive. Our manuscript is revised to highlight the importance of the material system, as you suggested. As design principles, the followings should be considered:

- 1) Adjusting the bandgap of the material to respond to the specific wavelength.
- 2) Ensuring that the materials exhibit a high intrinsic dipole moment upon excitation.
- 3) Preserving the high dipole moment with an extended lifetime, which can be achieved by designing the material to undergo the ESIPT process.
- 4) Maximizing photo-responsivity.

In detail, to highlight the importance of the material system, firstly, we present our general design principle as follows (page 4). *“For this purpose, the first step is to control the bandgap of the material to respond to a specific wavelength. Next, the materials should exhibit a high intrinsic dipole moment upon excitation, and the excited state lifetime should be extended to ensure the effectiveness of the excited state. In this section, we will focus on the design principles to control the bandgap with enhanced intrinsic dipole moment, while the design for extended lifetime at the excitation will be discussed in detail in the next section, along with the operational principle of the memristor.”*

Next, we explain the design principle for adjusting the bandgap and dipole moment as follows (page 4 and 5). *“Three different material design strategies—an enhancement of the donor-acceptor (D-A) strength, an extension of the effective conjugation length, and an introduction of nodal plane model—are used to synthesize the organic semiconductors having proper optical bandgaps and high dipole strengths at their excitation, which are applied as an interfacial layer of the memristor. To prepare UV-responding molecules, the dinaphthylamine is applied as an electron-donating moiety, and electron-deficient imidazole group is combined to control the highest occupied molecular orbital (HOMO) and lowest unoccupied molecular orbital (LUMO) energy levels of the molecules as D-A molecular structures (DNH, DN, DNH-F, and DN-F in Fig. 1a,e,b,f). DNH and DN were designed in our previous work, and DNH-F and DN-F, to which fluorine functional groups are added for the higher polarity (e.g. intrinsic dipole moment), are newly prepared to enhance the photo-responsivity of the device in the UV wavelength region. The asymmetric D-A structure is*

beneficial for improving intramolecular charge transfer characteristics of the molecule, which is favorable to provide high polarity, and the dinaphthyl groups in these molecules are also advantageous for extending the conjugation length for high intrinsic dipole moments.

However, the effective conjugation length of the imidazole-based system is not long enough to extend their absorption wavelength to visible region (emission in NIR). Therefore, as visible light-responding molecules, we prepared two NIR-emitting chalcone derivative fluorophores, CH-M and C-M (Fig. 1c,g), in which the hydroxyphenyl part of 2'-hydroxychalcone was replaced by hydroxynaphthyl group and naphthyl group to reduce bandgap of the molecules by introducing nodal-plane model and extending their effective conjugation length. The dimethylamine group is also beneficial to enhancing the D-A strength and further reducing their optical bandgap for covering the entire visible wavelength region by increasing its effective conjugation length. Moreover, to enhance the color-recognition ability of the photonic synapse device, CH-P and C-P (Fig. 1d,h) are further designed with additional modifications. The substitution of electron donating group at 4-position of phenyl group leads to an enhancement of donor-acceptor (D-A) ability of the molecule. Particularly, the dimethyl groups in parent molecules, CH-M and C-M, are replaced by two electron-rich phenyl groups, resulting in the formation of CH-P and C-P, respectively. We will discuss in detail how these molecules enhance the color-recognition ability of the device at a later time."

Third, we explain the design principle **for extending the lifetime upon excitation** as follows (page 7). "To make this photo-enhanced dipole effect more valid, we designed our molecules to have photo-tautomerization between enol (E) and keto (K) form at the excitation through intramolecular H-bond (e.g. -OH and -N=), which can follow the excited-state intramolecular proton transfer (ESIPT) process, consequently extending its lifetime at the photo-excitation. The ESIPT process is composed of four-level cyclic proton-transfer reactions (e.g. $E \rightarrow E^* \rightarrow K^* \rightarrow K \rightarrow E$; * symbol represents the excited state) (Supplementary Fig. 23), and this inherent four-level character is due to the stable K^* form in the excited state, leading the population inversion in K^* state. This inversion extends the effective lifetime of the photo-excited state as K^* form, which can make the photo-enhanced dipole moment effect more valid and thus effectively intensify the electric field within the device at the light-irradiation. Therefore, light can be utilized as an input signal of the memristor that can replace the electrical voltage bias for the conductance change of the device, even regardless of the type and switching mechanism of the medium.

Forth, we explain the design principle **for maximizing the photo-responsivity of the material to the photonic memristor** as follows (page 8 and 9). "Additionally, to further prove that the molecule with higher intrinsic dipole moment at the excitation is more effective on the conductance adjustment of the device, DNH-F (Fig. 1b) having fluorine functional groups for the higher polarity is designed, and DN-F (Fig. 1f), of which hydroxyl group is removed to be a non-proton-transfer molecule of DNH-F for comparison, is also prepared. Furthermore, to extend the light-responding functionality of the memristor to the visible region, a chalcone derivative fluorophore, CH-M (Fig. 1c), which can follow the ESIPT process at the visible light-irradiation, is prepared, and C-M (Fig. 1g), a non-ESIPT molecule of CH-M, is also synthesized for comparison. Finally, to optimize the spectral responsivity of the ESIPT molecule for realizing RGB-cognitive photonic memristor, CH-P (Fig. 1d), along with its non-ESIPT molecule C-P (Fig. 1h), are designed.

The dipole moment values of the designed molecules, along with their energy levels and frontier orbital distributions, are calculated by density functional theory (DFT) method. The orbital distributions of molecules at every energy state are displayed with their estimated intrinsic dipole moment values in Supplementary Figs. 24 and 25, and, especially, the orbital distributions and intrinsic dipole moment values at the excited state- K^* forms as for the ESIPT molecules (DNH, DNH-F, CH-M, and CH-P)—are represented in Fig. 1a-h. As designed, DNH-F (19.74 D) is confirmed to have much higher dipole moment than DNH (11.27 D) at the excitation. Meanwhile, at the ground state (E form) of CH-P, the molecular orbital mainly exists on electron-rich triphenylamine part of the molecule while that of CH-M is delocalized on entire molecule. Therefore, the substitution of dimethylamine (CH-M) to the strong donor diphenylamine (CH-P) could enhance the intramolecular charge transfer (ICT) characteristics."

- **Review comment 4:** *The application of this paper only shows that this synapse is used to construct the input layer for color recognition, while other network frameworks are existed, and it does not show the difference between this device and other synapses in other network structures except the input layer. It is difficult to reflect the uniqueness of this device in neural network.*

Our response: The photonic synapses designed in this work have **broader applications beyond serving as the input layer for color recognition, which is different from what reviewer commented.** While the CH-P-integrated memristor array is indeed used for constructing the input layer of the CNN for color discrimination as a pre-processing step, the DNH-F-integrated memristor array is responsible for composing **the convolutional kernels and fully-connected layers (FCLs)** of the CNN. As a result, we propose **a light-based intelligent system where the synaptic weights can be updated through optical stimulation.** This type of neural network system offers several advantages, including higher spatiotemporal resolution and a simplified weight-update methodology for the hardware array. Optical programming, in general, provides the potential for ultrafast speeds, low energy consumption, and large bandwidths by harnessing photons. Additionally, light irradiation can serve as an additional non-contact terminal for data writing, reducing power loss from electrical interconnects during electrical programming.

- **Review comment 5:** *More synaptic characteristics under photoelectric control and the proposal of a new architecture network based on these characteristic should be provided.*

Our response: Thank you for your comment to make our manuscript more constructive. To emphasize the uniqueness of our devices within a neural network framework, **we newly design a light-programmable reservoir computing system.** We confirm that our memristor can exhibit threshold switching characteristics at low voltage, which enables us to extend the application of our device to a light-programmable reservoir computing system. This extension showcases the distinctive features of our devices. We incorporated a new chapter titled "Extension to reservoir computing system" to provide detailed information on this neural network. Please refer to pages 21-23, Fig. 8, and Supplementary Fig. 37 for further details.

Fig. 8 **a** A schematic of the process flow in reservoir computing system, composed of input, reservoir, readout, and output layer. The number of pixels is minimized by encoding. **b** An original MNIST pattern consisting of 400 pixels (20×20) is compressed into a 100-pixel pattern (5×20) through RC layer. **c** Measured 20-cycle d.c. I - V characteristics of CH-P-integrated memristor at low voltages (I_{cc} was set to 0.11 mA) exhibiting threshold switching. **d** PSC variation of CH-P-integrated memristor at a single red-light (450 nm) pulse depending on the amplitude of pulse (1 – 20 mW cm^{-2}). Inset figure is large magnification. **e** PSC variation of CH-P-integrated memristor at four consecutive red-light pulses. Amplitude and interval of red-light pulses are 5 mW cm^{-2} and 50 ms. Pulse width in (d) and (e) is fixed at 50 ms. **f,g** PSC responses of CH-P-integrated memristor to 16 different cases of pulse stream. PSC data measured from 10 devices are presented in (g). Read voltage in (d-g) is 0.05 V. **h** Inference accuracy of the neural network with the light-programmable RC system.

- **Review comment 6:** Regarding the light enhancement part of the device, the mechanism is not solid and well discussed. For instance, simulation of the underlining mechanism is suggested to be supplemented.

Our response: Thank you for your comment on making our manuscript more comprehensive. To provide a more solid explanation of the underlying mechanism of the photonic memristor device, **we newly conducted a Mott-Schottky analysis under RGB light conditions for this revision** (Fig. 1o and page 7, 8). These results confirm that **light can increase built-in potential and electric field within the device** (from V_d and E_d to V_{dl} and E_{dl} , which are higher than V_d and E_d , respectively), as demonstrated in Fig. 1m,n and on page 6. This observation suggests that light can serve as an input signal instead of voltage.

Furthermore, through the Mott-Schottky analysis, we also prove that **the built-in potential (and thus electric field within the device) increases as the wavelength of the incident light decreases**, from red to blue (Fig. 1o and page 18). This forms the basis for the color-recognition ability of the device. We believe that these newly conducted analysis results provide direct evidence supporting our proposed mechanism for the photo-synaptic characteristics and color-recognition ability, thereby helping readers better understand the operation principles of the device.

Fig. 1 o Mott-Schottky plots of CH-P-integrated memristor (FTO/NiO/ organic interlayer/PMMA/Ag) at dark and under R-, G-, and B-light irradiation (450, 525, and 630 nm).

2. About reviewer #2's comments:

- We appreciate the reviewer's comment because we could have an opportunity to enhance color-distinction ability of our photonic synapse. The reviewer considered that *the artificial vision system RGB image encoding for pre-processing and hardware implementation of matrix multiplication programmable by optical flux attract great interests, while it is hardly convinced that this work holds promise for use in multispectral in-sensor vision computing due to multiple limitations* (as written in reviewer's comments).
 - 1) This opinion is mainly due to the reviewer's **misunderstanding that the wavelength-dependent photo-responsivity of the memristor solely relies on the absorbance of the molecule (density of the absorbed photons)**. However, it is important to note that the wavelength-dependent functionality also depends on the **lifetime of the molecule at the excitation, which varies with the wavelength**. We will further elaborate on this point in our point-to-point responses to provide a clearer explanation (answer to your comment 1, main page 17, and Supplementary Note 9).
 - 2) While it is true that the photo-responsivity is not solely determined by absorbance, we acknowledge that the CH-M-integrated memristor in the previous manuscript did not adequately demonstrate a clear distinction between RGB colors. Therefore, **we designed a new material CH-P and confirmed that a CH-P-integrated memristor can exhibit significantly enhanced color-discrimination ability**, compared to the CH-M-based memristor in the previous manuscript (Fig. 1, 2, 4, 5, and 6).
 - 3) The reviewer also mentioned that *some two-terminal optical memristors with multi-spectral (in visible spectra) responsivity have already been reported in other literature*. However, **these works were unable to fully demonstrate multi-spectral responsivity for color discrimination**, and they don't have the universality that our work have shown.

Followings are in-depth explanations for your comments.

- **Review summary:** *“The author has developed two-terminal optical memristors utilizing various combinations of metals and organic materials. Through experimentation, the author has claimed that they have demonstrated two memristors with distinct functionalities—a multispectral optical memristor for sensing and an optically programmable memristor for computing in a crossbar configuration. The artificial vision system RGB image encoding for pre-processing, and hardware implementation of matrix multiplication programmable by optical flux attract great interests, while it is hardly convinced that this work holds promise for use in multispectral in-sensor vision computing due to multiple limitations. The manuscript has a few shortcomings that need to be addressed. The manuscript contains redundant information with too complex sentences, and several portions of the methodology are not clear.”*

Our response: Thank you for your comment that our work on *“the artificial vision system RGB image encoding for pre-processing and hardware implementation of matrix multiplication programmable by optical flux”* has attract great interest. In this revision, we believe that all your concerns about *“the shortcomings that need to be addressed to hold promise for use in multispectral in-sensor vision computing”* are fully resolved. As you acknowledged, we have demonstrated two memristors with distinct functionalities. **Firstly**, as for *“the multispectral optical memristor for sensing in a crossbar configuration”*, **we newly design another material CH-P for this revision** (Fig. 1, page 26-27, Supplementary Fig. 2, 4, 17, 18, 19, 20, 22, 25, 26, and Supplementary Note 7, 8), **which can greatly enhance color-distinction characteristics of a multispectral optical memristor for sensing**. With the CH-P-integrated memristor, **we confirm that RGB colors, with various intensities ranging from 1 to 20 mWcm⁻²** (Fig. 6), **a light intensity range applicable to human visual systems without safety concerns** (*Health Phys.* 2013, 105, 74 / *Vision Res.* 2016, 121, 57 / *Adv. Mater.* 2022, 34, 2108979), **can be perfectly distinguished**. Secondly, as for *“the optically programmable memristor for computing in crossbar configuration”*, we address the advantages of optical programming over electrical programming of memristors used in the crossbar structure (page 20). Additionally, we have also reduced redundant information and clarified several portions of the methodology that you found unclear.

- **Review comment 1:** Authors claim that their device is “multispectral color-cognitive memristor” which is the main theme of this manuscript. It is clear that their molecule provides spectral resolution between UV and visible. However, it is hard to believe the color-cognition function in visible range since there is no spectral selectivity, but it only shows non-uniform broad absorption spectrum. Thus, this device would only work for color cognition under very specific conditions such as illumination of identical intensity RGB light with specific wavelength to the device simultaneously etc. For example, CH-M based device cannot distinguish between ~400nm and ~500nm wavelength light with identical intensity where the absorption coefficient is the same. And, it cannot distinguish 525nm wavelength light from 450nm wavelength light with half intensity of it since the absorption coefficient is half at 525nm. In my opinion, this is just memristor that can be programmed by visible light unless individual RGB sensors with color filters are integrated as pixel.

Our response: Thank you for your opinion to the color-cognition function of our photonic memristor. First of all, **we newly designed CH-P-material** (Fig. 1, page 26-27, Supplementary Fig. 2, 4, 17, 18, 19, 20, 22, 25, 26, and Supplementary Note 7, 8) and **CH-P-based memristor** (Fig. 4, 5, and 6) **for this revision. This new system provides spectral selectivity** to the memristor through the use of ESIPT molecule, even only with a **non-uniform absorption spectrum**.

In this work, **the ESIPT molecule thin film is not directly utilized as an active layer** that can provide charge carrier output current from the molecule itself (e.g., electron-hole pair generation by light absorption), as in a conventional photodiode, but applied **as a built-in potential enhancer that can increase the electric field**. The strong dipole effect of organic molecule at the excitation, enhancing this built-in potential of the device, is **more effective** when **the density of the excited molecules increases**, which is determined by **1) the density of the absorbed photon** and **2) their lifetime at the excitation**.

In our work, the **lifetime at the excitation could be extended** by designing the material to follow **ESIPT process**, which can extend its excited state lifetime through its stable Keto* form. Meanwhile, in general, **the excited state lifetime** of an organic semiconductor **undergoing ESIPT** can exhibit a trend where **it increases with decreasing wavelength** (or increasing energy) **of the excitation light**. This is because **higher-energy photons** (shorter wavelength light) can provide the necessary energy **to overcome the barrier** for efficient and rapid proton transfer processes **from Enol* to Keto***, leading to a stabilization of the excited state and a longer excited state lifetime—a more detailed discussion regarding this can be found in the end of this answer to your comment 1 and Supplementary Note 9. On the other hand, lower-energy photons (longer wavelength light) may not provide enough energy for efficient proton transfer processes, leading to a shorter excited state lifetime. The newly added Supplementary Fig. 34 confirms that our ESIPT molecules also follow the similar trend.

As mentioned, because **the number of excited molecules ($N_{excited}$)** is determined by **its lifetime (τ)** as well as **the number of the absorbed photon (N_{abs})** and follows an equation: $N_{excited} \sim N_{abs} \times (1 - \exp(-k\tau))$, where k is related to the area of the sample and absorption cross-section, **the output current of the device is not simply determined by the absorbance difference as reviewer pointed**. For example, even in the former CH-M-integrated memristor, the output current at 525 nm wavelength light cannot be around half of the output current at 450 nm wavelength light, simply judged by the absorbance difference.

Our goal in this work is to design a photonic memristor with the color-cognition function that can **discriminate light between 450 and 650 nm in visible range, providing RGB colors, under various intensity**. Although the density of the excited molecule is not simply determined by the absorbance and the lifetime of the excited molecule should be considered, we felt that the absorption gradient of the CH-M in the former manuscript is not enough to prove clear distinction between RGB light. Therefore, **we newly prepare CH-P**, which is designed to **enhance the photo-synaptic characteristics** and **color-discrimination ability** of the device, even **under different light power** conditions, **for this revision**. This is achieved by increasing its absorbance around 450 nm, resulting in a shaper absorbance curve that covers the visible wavelength range. As a result, the absorbance difference between 450 and 650 nm, which are the wavelengths targeted for discriminating RGB colors (Fig. 1j), is enhanced. Moreover, its PL lifetime decreases as the excited wavelength increases (Supplementary Figure 34b), further magnifying the gradient of the excited state dipole effect under each colored light.

As shown in Fig. 6d, **the output current values of the device under RGB colors, with various intensities ranging from 1 to 20 mWcm⁻²**, a light intensity range applicable to human visual systems without safety concerns systems (*Health Phys.* 2013, 105, 74, *Vision Res.* 2016, 121, 57, *Adv. Mater.* 2022, 34, 2108979), can be perfectly distinguished. For each RGB-light, the output current values under these various light intensities are in the range of 0.01 – 0.12 μA , 1.68 – 4.19 μA , and 104.05 – 210.37 μA , respectively (Fig. 6d,f), which **provide dynamic range values for color-distinction of 28–48 dB, 71–78 dB, and 106 – 112 dB, respectively (Fig. 6f), representing distinct color-distinguishable photo-synaptic characteristics**. The dynamic range for color-distinction is defined by the following equation: dynamic range = $20\log(\text{PSC}_{\text{max}}/\text{PSC}_{\text{min}})$, where PSC_{max} is steady state PSC after 50th pulse under 20 mW cm⁻² at each color, and PSC_{min} is steady state PSC after 1st pulse under 1 mW cm⁻² at red-light pulse.

We update all the figures to show the photo-synaptic functionality of the CH-P-based memristor, which represents a significant improvement over the former CH-M-based system (Fig. 1-6). This new material system demonstrates spectral selectivity in the memristor even only with a non-uniform absorbance, resulting in clear differentiation in wavelength selection coverage with various intensities, applicable to human visual systems. Our study successfully demonstrated perfect separation of RGB colors at specific wavelengths and under various light intensities. **By further optimizing the materials to maximize the gradient of absorbance and excitation lifetime**, it is expected that **even more precise wavelength selection in various intensity ranges can be achieved**. Thus, we believe that our photonic memristor presents a novel strategy for effectively emulating human retina-like color recognition systems.

Fig. 6 d PSC distribution depending on the number of pulses, obtained from CH-P-integrated memristor under RGB-light pulses with various light intensities from 1 to 20 mW cm⁻². **f** Dynamic range of CH-P-integrated memristor for color-distinction under various light intensities from 1 to 20 mW cm⁻².

Supplementary Fig. 34 Time-resolved photoluminescence of organic semiconductor thin films casted on quartz excited at 405 and 520 nm wavelengths. **a** CH-M (τ_{avg} : 3.16×10^{-10} s at 405 nm, 1.56×10^{-10} s at 520 nm). **b** CH-P (τ_{avg} : 3.89×10^{-10} s at 405 nm, 1.94×10^{-10} s at 520 nm).

About lifetime variation of ESIPT molecule depending on the excitation wavelength in detail:

[In Supplementary Note 9]

Luminescence spectra of ESIPT materials as a function of the excitation wavelengths of imidazole-based molecules or charcone-based materials are not extensively investigated so far. However, similar ESIPT systems including 6,6'-dimethyl3,3'-dihydroxy-2,2'-bipyridine (BP(OH)₂), 2-(20-hydroxyphenyl)benzothiazole (HBT), 2-(20-hydroxyphenyl)benzoxazole (HBO), and ortho-hydroxybenzaldehyde (OHBA) are reported to identify the relationships between the excitation wavelength and emission properties of the materials^{61,62}. For example, in (BP(OH)₂) molecule, emission from Franck-Condon excited state (the S₁ state) showed weaker fluorescence at 490 nm but the emission which are derived from higher excited states (S₂ or higher) exhibited more intense emission at 535 nm as a result of barrierless efficient ESIPT process⁶¹. Moreover, it was reported that the skeletal motions of the ESIPT molecules contributes to the reaction path of the proton transfer⁶². The UV-visible pump-probe spectroscopy for HBT, HBO, and OHBA with 30 femtosec resolution showed that a bending motion of the molecular skeleton reduces the proton donor-acceptor (D-A) distance, and the ESIPT process can only occur when sufficient vibrational energy is injected into the Franck-Condon state by the high-energy excitation⁶². Recent studies for the excitation wavelength and emission relationships on triazole derivatives showed a bright ESIPT luminescence upon high-energy excitation (less than 350 nm) with 16.5% PLQY, but relatively weak excimer fluorescence with 3.9% PLQY under low-energy irradiation longer than 365 nm⁶³. Considering that the PLQY is proportional to the fluorescence lifetime in a system which is free from dynamic collisional quenching, high-energy excitation for our imidazole- and charcone-based molecules could induce extended fluorescence lifetime, together with more efficient ESIPT after the photoexcitation.

[61] Ulrich, G., Nastasi, F., Retailleau, P., Puntoriero, F., Ziessel, R. & Campagna, S. Luminescent excited-state intramolecular proton-transfer (ESIPT) dyes based on 4-Alkyne-Functionalized [2,2'-Bipyridine]-3,3'-diol dyes. *Chem. Eur. J.* 14, 4381–4392 (2008).

[62] Lochbrunner, S., Stock, K., Riedle, E. Direct observation of the nuclear motion during ultrafast intramolecular proton transfer. *J. Mol. Struct.* 700, 13–18 (2004).

[63] Zhang, Y., Yang, H., Ma, H., Bian, G., Zang, Q., Sun, J., Zhang, C., An, Z. & Wong, W.-Y. Excitation wavelength dependent fluorescence of an ESIPT triazole derivative for amine sensing and anti-counterfeiting applications. *Angew. Chem. Int. Ed.* 58, 8773-8778 (2019).

- **Review comment 2:** *Some two-terminal optical memristors with multi-spectral (in visible spectra) responsivity have already been reported in other literature¹⁻³. What are the advantages of the proposed device over these references? Please consider these references in Supplementary Table. 1 as well as the ON/OFF ratio, which is a key metric to realize the effective memristor-based multiplication.*

1. Pei, Yifei, et al. "Artificial visual perception nervous system based on low-dimensional material photoelectric memristors." *ACS nano* 15.11 (2021): 17319-17326.

2. Wang, Wenxiao, et al. "Artificial optoelectronic synapses based on TiNxO2-x/MoS2 heterojunction for neuromorphic computing and visual system." *Advanced Functional Materials* 31.34 (2021): 2101201.

3. Hu, Lingxiang, et al. "All-optically controlled memristor for optoelectronic neuromorphic computing." *Advanced Functional Materials* 31.4 (2021): 2005582.

4. Chen, Pai-Yu, Xiaochen Peng, and Shimeng Yu. "NeuroSim: A circuit-level macro model for benchmarking neuro-inspired architectures in online learning." *IEEE Transactions on Computer-Aided Design of Integrated Circuits and Systems* 37.12 (2018): 3067-3080.

Our response: Thank you for your comment. We summarize the limitations of the references that you commented (1~3: *ACS nano* 2021, 15, 17319 / *Adv. Funct. Mater.* 2021, 31, 2101201 / *Adv. Funct. Mater.* 2021, 31, 2005582) in the following table. Now, they are included in Supplementary Table 1, and ON/OFF ratio data of all the work are newly added to Supplementary Table 1, as you suggested

References 2 and 3 demonstrated photonic memristors that responded to light of different wavelengths, but **they did not have multi-spectral responsivity for color discrimination.** Even, **they were already**

included in the Supplementary Table 1 of **the previous version of manuscript**. The memristor in **reference 1** was reported to respond to light of different wavelengths, but **the information about the wavelengths is not provided**, making it difficult to evaluate multi-spectral responsivity for color discrimination. In addition, photonic synapse properties such as **short-term plasticity (EPSC and PPF) and long-term plasticity were not investigated in detail**, and **the reported parameters were not competitive with our work**. Furthermore, as we have emphasized in our manuscript, most of works in the literature, including references 1-3, require specific material combinations to exhibit their photo-synaptic functionalities, limiting their integration into various systems. In contrast, **our work introduces a color-discriminating synaptic functionality into a two-terminal memristor, regardless of the switching medium**, which can **maximize the applicability of our work to existing technologies**. Meanwhile, **reference 4** is not about photonic memristor but rather NeuroSim program to provide circuit-level performance evaluation. Followings are details.

	This work	1	2	3
Photonic synapse properties	 - Distinct RGB color discrimination (450/525/630 nm) at different power (1~20 mW cm⁻²) - Short-term plasticity  ♦ EPSC behavior on each RGB studied ♦ PPF behavior on each RGB studied - Long-term plasticity  ♦ LTP/LTD at 600 pulses studied ♦ Nonlinearity: 0.5/0.7 ♦ Symmetricity: 54 	 - Conductance variation at 3 different light conditions (but no wavelength information is provided) - Short-term plasticity  ♦ EPSC behavior: not studied ♦ PPF behavior: not studied - Long-term plasticity  ♦ LTP only at 10 pulses studied ♦ Nonlinearity: 4.3/x ♦ Symmetricity: x 	 - No multi-spectral responsivity for color discrimination: All photonic synapse properties were studied using UV light. Although the EPSC behaviors depending on RGB light conditions were presented in Fig. S13, the aim was only to highlight the importance of UV light due to their weak signals that were insufficient for LTP, as explained in the paper. 	 - No multi-spectral responsivity for color discrimination: While it has been demonstrated that memristors can produce various levels of current depending on the wavelength of light, this feature is not intended for color discrimination in vision systems. Rather, it is used to switch SET and RESET behavior of the device. (e.g., blue: LTP and NIR: LTD).
Memristor reliability	1.1×10 ⁵ cycles	~ 10 ⁵ cycles	-	-
Medium	Free of medium	PbS quantum dots	TiN _x O _{2-x} /MoS ₂	O _D -IGZO/O _R -IGZO
Application to neural network	CNN & Reservoir computing	No (neuron circuit with other diffusive memristor)	No	No

- **Review comment 3:** Figure 3j shows the retention time of CH-M at LRS only, thus the author needs to include more information above to claim the simulated memristor crossbar based on the DNH-F memristors. What are the retention times of both CH-M and DNH-F memristors at HRS and LRS, respectively?

Our response: Thank you for your comment to make our manuscript more comprehensive. As you suggested, to claim the simulated crossbar arrays based on CH-P or DNH-F memristors, we **newly added the retention behaviors of both CH-P- and DNH-F-integrated memristors at both LRS and HRS** (Fig. 3j).

Fig. 3 j Retention test results of DNH-F and CH-P-integrated memristors at LRS [after 50 consecutive blue light pulses (450 nm, power: 5 mW cm⁻², width: 1 s, and interval: 0.5 s)] and HRS. Read voltage 0.05 V is applied during test.

- **Review comment 4:** *The author has provided various device structures with different material combinations, some of which may be redundant in guiding readers to the conclusion. Since the artificial vision system includes only CH-M and DNH-F memristors, other device structures and material combinations can be regarded as optimization/improvement procedures for the ultimate artificial vision system. It is recommended that the main figures be rearranged, and the corresponding texts revised accordingly, considering the ultimate application and the potential universal applicability of the device.*

Our response: Thank you for your feedback to help us make our manuscript more reader-friendly. As per your suggestion, **we rearranged most of the main figures** related to the memristor devices and revised the corresponding texts accordingly. **We moved all the plots about DNH and CH-M memristors,** regarded as optimization/improvement procedures for the ultimate artificial vision system, **to Supplementary Figures.**

- **Review comment 5:** *The advantages of optical programming mentioned on page 17 may not be applicable to this work, as the corresponding ref. 19 involves a three-terminal device with a 2D material. The author should consider providing other advantages of optical programming over electrical programming of memristors (DNH-F) used in the crossbar structure. It is also challenging to focus the optical beam in a scaled-down memristor crossbar in nanometers to selectively program each memristor pixel, and electrical programming allows for nanosecond programming capability with compact electrical circuitry to minimize parasitic capacitance and inductance.*

Our response: Thank you for your feedback on making our manuscript more comprehensive. The **high spatiotemporal resolution** can be achieved by the optical programming, because light exposure can replace the application of electrical bias, thereby reducing the need for a contacting terminal. And the **simplified weight update methodology** is feasible because light can be directly exposed to all the targeted memristor cells at the same time, while electrical programming requires complicated combinations of voltage pulses between bottom and top electrodes through columns and rows that can update all the targeted memristor cells step by step. Therefore, **the advantages of optical programming mentioned on page 20** (formerly page 17), such as higher spatiotemporal resolution and simplified weight update methodology, **are not limited to the 3-terminal device with a 2D material** but are also applicable to our 2-terminal structure.

If you thought optical programming was not applicable to our work because a 3-terminal device with a 2D material in reference 20 (formerly reference 19) has a window area that can receive light (while our memristor is enclosed by top and bottom electrodes), this is not correct because our memristor has a transparent electrode, which makes the window area used for the 3-terminal device-based crossbar array in reference 20 unnecessary. This makes our 2-terminal architecture even more advantageous for high integration density.

To further support, we newly included additional advantages of optical programming over electrical programming of memristors in response to your suggestion (page 20). Optical programming offers **the potential for ultrafast speeds, low energy consumption, and large bandwidths by utilizing photons** (*Phys. Status Solidi RRL*. 2019, 13, 1900082). Furthermore light-irradiation can serve as **an additional non-contacting terminal for data-writing** that can **reduce power loss** from electrical interconnects during the electrical programming (*Light Sci. Appl.* 2019, 8, 42).

Meanwhile, regarding the optical beam focusing, the physical limitation of diffraction has made it challenging to achieve downsizing and high integration density in optical programming, as you mentioned. However, in the past decade, there have been attempts to extend advantages of optical technology in terms of speed and energy to memory field through the rapid development of optical integration technology, and significant progress has been made in performance, functionality, and operational aspects of bit-level storage (*Light. Sci. Appl.* 2020, 9, 91). As part of these efforts, recent works have actively investigated **the use of optical waveguides to deliver light to nanoscale devices** (*Sci. Adv.* 2019, 5, 2687 / *Nano Lett.* 2021, 12, 5784 / *Nanoscale. Res. Lett.* 2007, 2, 219). A waveguide focuses and transmits light in a specific direction, making it useful for concentrating and delivering optical signals. These approaches have even been **proposed in neuromorphic computing** to selectively input optical signals into nanoscale components, providing faster programming speed and lower energy consumption than electrical programming.

- **Review comment 6:** Author explained that every color input image is separated into RGB three channels through the CH-M-integrated memristor array as a pre-processing. However, it is almost impossible to perform this task using the demonstrated device since the image color is super-positioned spectrum that cannot be resolved using the absorption spectrum of CH-M based memristor. For example, if Red and Green light is illuminated together, then how this single device can recognize the color without color filter? If there is any mechanism to achieve this using the proposed device, it should be clearly explained.

Our response: Thank you for your comment to make our manuscript more comprehensive. As we answer to your review comment 1 in detail, we newly design another material CH-P for this revision, and the CH-P-integrated memristor confirm that RGB colors, with various intensities ranging from 1 to 20 mWcm⁻², a light intensity range applicable to human visual systems without safety concerns, can be perfectly distinguished. As we explained, the strong dipole effect of organic molecule at the excitation, enhancing the built-in potential of the device, is more effective when the density of the excited molecules increases, which is determined by 1) the density of the absorbed photon and 2) their lifetime at the excitation. Both of these factors are wavelength-dependent. The detailed mechanism to achieve the color-discrimination ability is explained in the answer to your review comment 1, page 17 in main text, and Supplementary Note 9. Consequently, for each RGB-light, the output current values under these various light intensities are in the range of 0.01 – 0.12 μA, 1.68 – 4.19 μA, and 104.05 – 210.37 μA, respectively (Fig. 6d,f), which provide dynamic range values for color-distinction of 28–48 dB, 71–78 dB, and 106–112 dB, respectively (Fig. 6f), representing distinct color-distinguishable photo-synaptic characteristics without color filter.

- **Review comment 7:** Authors claimed that the synaptic weight of each cell is defined by the conductance difference between two equivalent DNH-F devices to represent negative weights. Did author employ negative weight value for the neural network simulation? If so, how can you generate the negative weight value just using two memristors? Based on author’s schematic description in figure 7d, it is not possible to create negative weight values at each pixels, but it shows summation at the column and process it through differential amplifier. This should be clearly explained.

Our response: Thank you for your comment. In software-based artificial neural network (ANN), both positive and negative values are utilized for the weight values of individual cells. However, in hardware crossbar array ANN, the conductance of the memristor (G), which represents the synaptic weight (W), cannot have negative values. Therefore, to demonstrate both positive and negative weight values in hardware ANNs, the synaptic weight of each cell is represented by the conductance difference ($W = G^+ - G^-$) of a pair of two equivalent synapses, of which the conductance values are represented by G^+ and G^- ($G^+ > 0$ and $G^- > 0$), as shown in the dashed red squares in the following figure. In this configuration, the total current (I) through an output node is represented by $I = I^+ - I^-$ ($I^+ > 0$ and $I^- > 0$). This can be further expressed as $I = I^+ - I^- = \sum_{i=1}^{1,024} V_i^{in} W_i = \sum_{i=1}^{1,024} V_i^{in} (G_i^+ - G_i^-) = \sum_{i=1}^{1,024} (V_i^{in} G_i^+ - V_i^{in} G_i^-)$. To help readers better understand this concept, we added new sentences and references (on page 20 and 30).

[61] Alibart, F., Zamanidoost, E., Strukov, D. B. Pattern classification by memristive crossbar circuits using *ex situ* and *in situ* training. *Nat. Commun.* 2013, 4, 2072.

[62] Kataeva, I., Merrikh-Bayat, F., Zamanidoost, E. & Strukov, D. Efficient training algorithms for neural networks based on memristive crossbar circuits. In *Proc. Int. Joint Conf. Neural Networks* 1–8 (IEEE, 2015).

- **Review comment 8:** What are G_{max} and G_{min} used in the NeuroSim simulation? Is the base experimental LTP/LTD (Supplementary Fig. 26a) the same result with Fig. 5h? The ON/OFF ratio of Fig. 5h is insufficient to realize the memristor-based matrix multiplication as mentioned in the NeuroSim paper 1.

I. Chen, Pai-Yu, Xiaochen Peng, and Shimeng Yu. "NeuroSim: A circuit-level macro model for benchmarking neuro-inspired architectures in online learning." *IEEE Transactions on Computer-Aided Design of Integrated Circuits and Systems* 37.12 (2018): 3067-3080.

Our response: Thank you for your questions to make our manuscript more comprehensive. The **G_{max} and G_{min} values** used in the NeuroSim simulation **are based on experimental LTP/LTD data**, consistent with the result in Fig. 5 and Supplementary Fig. 33, as you mentioned. The reason, why the device presented in **Fig. 15b of the reference** *IEEE Trans. Comput.-Aided Des. Integr. Circuits Syst.* 2018, 37, 3067 requires **high ON/OFF ratio** (dynamic ratio) (>10) **to achieve high accuracy**, is because **the calculations in that reference are based on the case with a small number of pulses (64/64, 6-bit, for LTD/LTD)**, which corresponds to the number of conductance states. In contrast, **our case involves a high number of pulses (300/300 for LTD/LTD)**. In the table presented below, we provide a summary of the accuracy variation in our neural network system based on the ON/OFF ratio (ranging from 1.4 to 140) and the number of pulses (ranging from 32 to 300). The non-linearity values of LTP and LTD are fixed at 1.1. As shown in the table, in the case of 32 pulses, the accuracy of the system significantly increases with the ON/OFF ratio from 69 % (ON/OFF: 1.4) to 86 % (ON/OFF: 140), similar to the data in the reference that you quoted. However, when the number of pulses is increased to 300, the system already achieves a high accuracy of 90 % at the ON/OFF ratio of 1.4, with a slightly improvement to 91 % at the ON/OFF ratio of 140.

	Non-linearity: 1.1 (P) / 1.1 (D)		
	On/Off 1.4	On/Off 14	On/Off 140
Pulse number 32	69 %	85 %	86 %
Pulse number 150	85 %	-	-
Pulse number 300	90 %	90 %	91 %

Additionally, non-linearity is also another crucial factor that affects the accuracy of the system. For reference, we also provide a summary of the accuracy variation in our neural network system based on the non-linearity values (ranging from 1.1 to 0.01) for both the 32-pulse and 300-pulse cases, with a fixed ON/OFF ratio of 1.4. The results are presented in the following table. As shown, at the ON/OFF ratio of 1.4, improving the non-linearity from 1.1 to 0.01 enhances the accuracy of the system from 69 % to 86 % in the 32-pulse case and from 90 % to 93 % in the 300-pulse case.

	On/Off 1.4		
	Non-linearity: 1.1 (P) / 1.1 (D)	Non-linearity: 0.1 (P) / 0.1 (D)	Non-linearity: 0.01 (P) / 0.01 (D)
Pulse number 32	69 %	80 %	86 %
Pulse number 300	90 %	92 %	93 %

Consequently, we confirm that various factors, such as the number of conductance states and non-linearity, are also crucial for improving the accuracy of neural network system in addition to the ON/OFF ratios. In particular, **our system has a high number of conductance states and low non-linearity, and thus we can achieve high accuracy even with a relatively low dynamic range (ON/OFF ratio).**

Review comment 9: *In method section, there is no description about characterization. Organic device is usually unstable under ambient condition. Is this device measured under ambient condition or under nitrogen condition? If it is measured in the glove box, how does it works under the ambient conditions?*

Our response: Thank you for your comment. **All measurements were conducted in ambient air, and we added all the information related to the characterization in method section** (page 28, 29), as follows:

Device measurement: All the measurement were conducted inside a dark shield box under ambient air condition. For the characterization of resistive switching and synaptic behaviors, d.c. I-V characteristics, EPSC, PPF, and LTP/LTD were measured using a probe station equipped with a Keithley 2636B semiconductor parameter analyzer. I-V characteristic measurements were conducted using two methods: double sweep mode (a unidirectional cyclic loop I-V measurement system, possible in either positive or negative direction) and cycle sweep mode (a bidirectional cyclic loop I-V measurement system). Additionally, an appropriate compliance current (15 mA was set for I-V cycle sweep measurement) was set to prevent device breakdown and the formation of permanent filaments. Synaptic behavior measurements were performed using the SP-300 model potentiostat (BioLogic) to generate optical pulses, and the behavior induced by the pulses was monitored in real-time by applying a constant read bias of 0.05 V in the time domain. Optical measurements were carried out by irradiating the optical beam onto the bottom electrode, FTO, of the device. The light sources for the optical pulses were light emitting diodes (LEDs), with available wavelengths of 365 nm (UV), 450 nm (blue), 525 nm (green), and 630 nm (red). The power of the light was calibrated using a photodiode sensor, the PD300 (Ophir), with ranges of 1 to 3 mW cm⁻² for UV and 1 to 20 mW cm⁻² for RGB. To characterize a single device, light is exposed through an aperture (1 mm diameter). For the RGB letter pattern, optical masks are used. The memristor pixels in a letter with the same color were illuminated simultaneously, and their output currents were sequentially read. Retention measurements were conducted on a hotplate, in ambient air environment, and within a dark shield box where light was blocked. The retention behaviors at both HRS and LRS were measured. Retention at LRS was taken into two conditions: after 50 blue light pulses and UV light pulses. Cycle endurance was conducted in 50-cycle segments. This was to prevent communication buffer issues between the computer and the equipment (Keithley 2636B) during long-term measurements. The transfer curve (V_{GS} - I_D) of FET was measured in two conditions: under light irradiation and dark condition. The dark condition was first conducted, and then light irradiation was performed. The light exposure lasted from the beginning to the end of the measurement and was maintained at a constant intensity.

- **Review comment 10:** *Below are the minor comments.*

1. *Please clarify and the terminologies, EPSC and PSC that were randomly used in the manuscript.*

Our response: Thank you for your comment. EPSC stands for "excitatory postsynaptic current response" and refers to the behavior of the device representing the increase in output current in response to an input signal, such as light in this work. PSC (postsynaptic current), on the other hand, refers to the output current itself from the synaptic device. These terminologies are used in their proper positions.

For reference, in neuroscience, an excitatory postsynaptic potential (EPSP) is a postsynaptic potential that increases the probability of a postsynaptic neuron firing an action potential. The flow of ions that causes an EPSP is known as an excitatory postsynaptic current (EPSC), and it exhibits short-term plasticity with volatile characteristics. The opposite of EPSPs are inhibitory postsynaptic potentials (IPSPs) and the corresponding current is referred to as an inhibitory postsynaptic current (IPSC).

2. *There are too many run-on and long sentences that are hard to be understood. Please split each information and simplify the sentences to improve the readability.*

Our response: Thank you for your comment. We corrected manuscript to improve the readability.

3. *There is typo in page 9 "OEFT"*

Our response: Thank you for your comment. We corrected the typo in page 10 (former 9) to OFET.

3. About reviewer #3's comments:

- **Review summary:** In this manuscript, the authors reported that the introduction of organic semiconductor layer in an NiO-based semi-transparent memristor, which has a long lasting strong molecular polarity in the photo-excited state. Some comments are as follows-

Our response: Thank you for your constructive comment on our manuscript to make our manuscript more comprehensive.

- **Review comment 1:** The authors reported that the bipolar resistive switching (BRS) in their NiO based-memristor is possible because of migration of “Ag-ions” (active electrode), whereas the device with the inert electrode “Au” does not exhibit BRS. But if we compare the compliance currents (CC) in Fig. 3e and supplementary Fig. 22, then we observed a large difference (\square 1 million times) between them. Is that possible if we apply the same amount (higher) of CC then BRS may be occurred in Au-based device?.

Our response: Thank you for your comment. We have already applied **the same compliance current (CC) of 1.5×10^{-2} A for both Ag-electrode and Au-electrode memristors** (Fig. 3e and Supplementary Fig. 30). The CC is set to prevent permanent device breakdown due to a large amount of current. However, the output current of the Au-electrode memristor is extremely small even at 5 V, which is much higher than the voltage applied to the Ag-electrode memristor (under 1 V) that reached the CC. The resistance of the Au-electrode memristor is too high to reach 1.5×10^{-2} A due to difficulty in filament formation. Therefore, setting the CC to be 1.5×10^{-2} A when measuring the Au-electrode memristor does not affect any of its properties.

Fig. 3 e Measured 10-cycle d.c. I - V characteristics of the memristor after electro-forming built with various organic thin films. Positive voltage bias is first applied to top Ag electrode (FTO: grounded): (e) log scale.

Supplementary Fig. 30 Measured 10-cycle d.c. I - V characteristics of the memristor built with Au top electrode (glass/FTO/NiO/PMMA/Au). Positive voltage bias is first applied to top Au electrode (FTO: grounded). **a** Normal scale. **b** Log scale.

- **Review comment 2:** *In supplementary table-1, the authors compared the non-linearity (NL) and symmetricity of the devices reported in literature, which have been estimated by extracting LTP/LTD data. I suggest that NOT to put those references in which NLs of both LTP and LTD curves did not mention (such as ref. 55, 56, 60, 65 etc.). Also, I am just curious and want to confirm that the symmetricity equation (supplementary note 7) will be precisely applicable in those cases (supplementary ref. 13, 52) where the number of applied pulses are different for LTP/LTD or not?*

Our response: Thank you for your comment on making our literature survey more convincing and accurate. As you suggested, we have removed all the supplementary references (33, 38, 55, 56, 60, 65 from the previous version) where the nonlinearity values for both LTP and LTD curves were not mentioned. The equation for symmetricity (Supplementary Note 10: Supplementary Note 7 in previous manuscript) is only applicable in cases where the number of applied pulses is the same for both LTP and LTD. We also removed the estimated symmetry values for supplementary references 13 and 52, as per your suggestion.

- **Review comment 3:** *In the section of 'Emulation of human visual system', the authors reported that the patterning of RGB-colored letters is illustrated by preparing a photonic synaptic array of 8×8 pixels. Is this an image array or memristive device array? In 'device fabrication' section, the device array preparation has not mentioned. Also, I will suggest the authors to specify the annealing environment for NiO layer.*

Our response: Thank you for your constructive comment. **The patterning of RGB-colored letters is illustrated by preparing a memristive device array.** We added a picture of memristive device array (Fig. 6e). We also included the device array preparation procedures in Method section (page 27-28). The annealing environment for NiO layer is also described in this section. Followings are details:

Crossbar array fabrication: FTO on glass substrate was used as bottom transparent electrode. To prepare a crossbar array, FTO-coated glass substrate underwent cleaning with acetone and IPA before being spin-coated with the solution of photoresist (PR) AZ5214 at a speed of 1000 rpm for 1 min. The deposited PR film was then subjected to a soft bake at 100 °C for 5 min. A patterned mask with 8 lines, of which line widths were 100 μm , was placed on top of the PR film, which was then exposed to UV light for 7 s. After exposure, the PR film was developed for 1 min in AZ 300 MIF solution, rinsed with DI water, and subjected to a hard bake at 120 °C for 5 min. The patterned positive PR-coated FTO film was etched for 20 min using CH_4 gas in a reactive ion etcher (with a gas flow rate of 90 sccm and reflective power of 300 W). The residual PR was removed using acetone and IPA. Next, the NiO thin film layer was deposited onto the patterned FTO substrate using spin-coating at 2000 rpm for 30 seconds, followed by annealing at 550°C for 2 hours in a furnace with ambient air. Organic molecules (DNH, DN, DNH-F, DN-F, CH-M, C-M, CH-P, and C-P) were dissolved in chlorobenzene and coated on the NiO layer (2,000 rpm for 30 s) and annealed on the hotplate at 100 °C for 10 min. Finally, Ag top electrode was deposited using a shadow mask with 8 lines, of which line widths were 100 μm , by thermal evaporator in a vacuum environment (less than 1×10^{-6} Torr).

Fig. 6e Optical microscopy image of 8×8 pixels of CH-P-integrated memristor array.

- **Review comment 4:** *The authors must provide the JCPDS file number (as a reference) to match the intensity and position of diffraction peaks in XRD pattern.*

Our response: Thank you for your constructive advice. We added the JCPDS file number as a reference to page 11 (Memristor device section) of the manuscript. *The X-ray diffraction (XRD) patterns of the spin-cast NiO thin film (Fig. 3a) show strong peaks at $2\theta = 37.5^\circ, 43.3^\circ, 62.9^\circ, 75.5^\circ,$ and $79.6^\circ,$ which correspond to (111), (200), (220), (311), and (222) planes of the cubic crystal structure (JCPDS file no. 65-5745), indicating that the synthesized NPs have a distinct crystalline structure even without any post-process.*

- **Review comment 5:** *If the fabricated memristor is not forming-free, then the authors must provide the required forming (the very first set) process in Fig. 3e.*

Our response: Thank you for your comment to make our manuscript more comprehensive. As you suggested, we newly added *I-V* characteristics of each device with and without the organic layer (DNH, DNH-F, CH-M, and CH-P) during forming process to the Supplementary Fig. 29.

Supplementary Fig. 29 Measured 1st and 11th cycle d.c. *I-V* characteristics of the memristor with different organic semiconductor (glass/FTO/NiO/PMMA/orgnic layer/Ag). Positive voltage bias is applied to top Ag electrode (FTO: grounded). **a** Without organic layer. **b-e** With organic layer (**b**: DNH, **c**: DNH-F, **d**: CH-M, and **e**: CH-P).

- **Review comment 6:** *The authors must mention the calculated values of relaxation times of PPF-index curve during the fast and slow phase of exponential decay.*

Our response: Thank you for your comment to make our manuscript more comprehensive. As you suggested, we newly added the calculated values of relaxation times of PPF-index curve during the fast and slow phase of exponential decay to the Supplementary Table 3 and main page 15.

Supplementary Table 3 Fitted parameters of PPF index.

	C_1	τ_1 (s)	C_2	τ_2 (s)
DNH	0.2133	0.39	0.1256	44.04
DNH-F	0.4036	0.53	0.1292	46.60
CH-M	0.1734	1.03	0.1186	69.58
CH-P	0.2349	0.27	0.8402	45.37

$$\text{where, PPF index}(\Delta t) = C_1 \exp(-\Delta t/\tau_1) + C_2 \exp(-\Delta t/\tau_2)$$

- **Review comment 7:** *The authors must mention the size of ‘laser spot area on device’, which was fixed during the optical measurements.*

Our response: Thank you for your comment to make our manuscript more comprehensive. We used light emitting diodes (LEDs) as light sources for the optical pulses. To characterize a single device, light is exposed through an aperture (1 mm diameter). For the RGB letter pattern, optical masks are used. The memristor pixels in a letter with the same color were illuminated simultaneously, and their output currents were sequentially read. We added this information to page 28 in “Device measurement” section of “Method”.

- **Review comment 8:** *There are some spelling mistakes in p.11.*

Our response: Thank you for your comment. We have corrected the typos in page 13 (paper 11 in previous version) you pointed out. “*The temporal cycle-to-cycle variation (CCV) results of the organic interlayer-integrated memristor confirm that it does not show any noticeable degradation of HRS-LRS ratio (Fig. 3g), I-V characteristics (Fig. 3h), and set voltage (Fig. 3i) even after 1.1×10^5 sweeping cycles, representing the robustness of the organic interlayer.*”

REVIEWER COMMENTS

Reviewer #1 (Remarks to the Author):

The authors have well addressed my concerns and this work can be accepted now.

Reviewer #2 (Remarks to the Author):

The author has put forth effort in offering additional information to address the queries raised by the reviewers, particularly fabricating a new CH-P-based photonics memristor. Most of the revisions focused on enhancing the clarity of the experimental methodology and the neuromorphic metrics pertaining to the device. Nevertheless, to emphasize the device's performance and potential, the color-distinguish capability of the proposed memristors is still not clear.

1. The author has presented an improved color-distinguishing protocol that incorporates the wavelength-dependent responsivity and relaxation time of the newly fabricated CH-P memristors. However, the term "color-distinguish" used in the manuscript should be applicable to practical scenarios, such as RGB image sensors or general color-distinguishing tests for color blindness. Specifically, each color can be represented as 8-bit (0~255) RGB-channel intensities, i.e, arbitrary color $X = (R, G, B) = (128, 65, 10)$. For the pre-processing RGB separation shown in Supplementary fig. 36, the best-performing CH-P memristor needs to separate the RGB channels with their analog states (0~255). However, Figure 6g only utilized monochromatic lights that encompass three colors: $X'H' = (255, 0, 0)$, $X'I' = (0, 0, 255)$, and $X'U' = (0, 255, 0)$, which significantly differs from the airplane image in Supplementary figure 36, containing numerous color combinations (0~255, 0~255, 0~255). According to Fig. 6, the proposed photonic memristors are only capable of distinguishing monochromatic colors, for example, $XRed = (0~255, 0, 0)$, $XGreen = (0, 0~255, 0)$, and $XBlue = (0, 0, 0~255)$. Due to the significantly higher responsivity of the blue wavelength compared to other colors, it is expected that the proposed photonic memristors will hardly distinguish some color combinations dominated by XBlue intensity, for example, (200, 255, 10) and (50, 0, 30).

Since the proposed CH-P photonic memristors do not achieve the ideal RGB channel separation, it is challenging to claim the pre-processing capability of the device as shown in Supplementary fig. 36 and Fig. 7. For the relaxation time differences, the wavelength-dependent relaxation time difference was also observed in other heterostructure, which is not the sole property of the proposed device.

If the CH-P device is capable of distinguishing RGB channels with multiple states (0~255, 0~255, 0~255), could the author demonstrate the PSC response of the device under various color combinations, i.e, R with '100' intensity + B with '10' intensity + G with '230' intensity? Note that the numbers ('100', '10', and '230') are relative values and do not necessarily need to be 8-bit. After illumination, the device should be able to distinguish each RGB channel separately as (100, 230, 10) based on the spectral responsivity and spectral relaxation time of the CH-P. Alternatively, the author could simulate the distinguishing capability based on the proposed spectral responsivity and relaxation time differences of the device. What potential bit resolution could be achieved?

In this regard, the term "color-distinguish" also needs to be reconsidered in revised Supplementary table. 1 and overall manuscript.

2. The author suggests that negative weight values can be achieved by utilizing a pair of memristors with comparators. By incorporating the proposed optical programming, each pixel containing two photonic memristors can represent both positive and negative weight values. Since a general weight matrix exhibits high intensity contrast between neighboring pixels (resulting in a noisy image when plotted), each programming beam needs to access every pixel with various intensities. The author proposed waveguide and plasmonic approaches as alternatives to achieve this, which would require microfabrication to pattern the waveguide and non-contact terminals in proximity, similar to the process of patterning metal lines for electrical programming.

Regarding the "high spatiotemporal resolution" capability of optical programming, it is important to note that according to Reference 20 in the manuscript, this term is applicable primarily to biological neurons, such as in optogenetics. In the case of solid-state semiconductor circuitry, however, electrical programming still provides a reliable means of quantization to represent multiple bits, particularly for memristors. Recent memristor crossbars have achieved 2048 states by employing electrical programming protocols.[1]

[1] Rao, Mingyi, et al. "Thousands of conductance levels in memristors integrated on CMOS." *Nature* 615.7954 (2023): 823-829.

3. Considering the uncertainty of the color-distinguish capability of the proposed device, it is not possible to clearly pre-process a full 8-bit RGB image at the current stage of development. Therefore,

it is recommended to compare the classification results using different approaches. One approach would involve partial preprocessing based on the requiring simulated device limitations, allowing for a certain percentage of overlap. The other approach would involve worse percentage overlaps purely in software.

4. The author has dedicated significant efforts to describing the experimental methodology. One original question was the stability of the fabricated organic devices. It is crucial to understand whether these devices exhibit stability under ambient condition. Additionally, information regarding the lifetimes of the organic devices in the ambient condition would be valuable to ascertain.

5. The author has made the revised manuscript more readable by addressing run-on and lengthy sentences. However, there are still a few instances where sentences contain an excessive number of phrases connected by multiple commas and "and." Therefore, it is recommended to split the information in these complex sentences for better clarity.

Followings are detailed responses to reviewer's comments:

Reviewer #2

Review summary: *The author has put forth effort in offering additional information to address the queries raised by the reviewers, particularly fabricating a new CH-P-based photonics memristor. Most of the revisions focused on enhancing the clarity of the experimental methodology and the neuromorphic metrics pertaining to the device. Nevertheless, to emphasize the device's performance and potential, the color-distinguish capability of the proposed memristors is still not clear.*

Our response: Thank you for your valuable comment. In this response, we provide comprehensive explanations of the color-distinguishing capability of the proposed memristors. We are confident that this clarification will effectively elucidate the color distinction characteristics of the synapse device.

Comment #1: *The author has presented an improved color-distinguishing protocol that incorporates the wavelength-dependent responsivity and relaxation time of the newly fabricated CH-P memristors. However, the term "color-distinguish" used in the manuscript should be applicable to practical scenarios, such as RGB image sensors or general color-distinguishing tests for color blindness. Specifically, each color can be represented as 8-bit (0~255) RGB-channel intensities, i.e, arbitrary color $X = (R, G, B) = (128, 65, 10)$. For the pre-processing RGB separation shown in Supplementary fig. 36, the best-performing CH-P memristor needs to separate the RGB channels with their analog states (0~255). However, Figure 6g only utilized monochromatic lights that encompass three colors: $X'H' = (255, 0, 0)$, $X'I' = (0, 0, 255)$, and $X'U' = (0, 255, 0)$, which significantly differs from the airplane image in Supplementary figure 36, containing numerous color combinations (0~255, 0~255, 0~255). According to Fig. 6, the proposed photonic memristors are only capable of distinguishing monochromatic colors, for example, $XRed = (0~255, 0, 0)$, $XGreen = (0, 0~255, 0)$, and $XBlue = (0, 0, 0~255)$. Due to the significantly higher responsivity of the blue wavelength compared to other colors, it is expected that the proposed photonic memristors will hardly distinguish some color combinations dominated by $XBlue$ intensity, for example, $(200, 255, 10)$ and $(50, 0, 30)$.*

Since the proposed CH-P photonic memristors do not achieve the ideal RGB channel separation, it is challenging to claim the pre-processing capability of the device as shown in Supplementary fig. 36 and Fig. 7. For the relaxation time differences, the wavelength-dependent relaxation time difference was also observed in other heterostructure, which is not the sole property of the proposed device.

If the CH-P device is capable of distinguishing RGB channels with multiple states (0~255, 0~255, 0~255), could the author demonstrate the PSC response of the device under various color combinations, i.e, R with '100' intensity + B with '10' intensity + G with '230' intensity? Note that the numbers ('100', '10', and '230') are relative values and do not necessarily need to be 8-bit. After illumination, the device should be able to distinguish each RGB channel separately as (100, 230, 10) based on the spectral responsivity and spectral relaxation time of the CH-P. Alternatively, the author could simulate the distinguishing capability based on the proposed spectral responsivity and relaxation time differences of the device. What potential bit resolution could be achieved?

In this regard, the term "color-distinguish" also needs to be reconsidered in revised Supplementary table. 1 and overall manuscript.

Our response: Thank you for your valuable comment, which has allowed us to present a more comprehensive color distinction mechanism. Contrary to the reviewer's comment, the CH-P memristor is not incapable of distinguishing RGB channels with multiple states. It not only responds to three individual RGB monochromatic lights, as demonstrated in Fig. 6g, but also effectively detects various color combinations (i.e., separates the RGB channels with their analog states) for the pre-processing in Supplementary Fig. 36 and Fig. 7. This "color-distinguish" capability, is achieved by satisfying the following three key characteristics of the photonic memristor.

Condition 1. The post-synaptic current (PSC) range of each RGB color must be adequately separated to avoid any overlap. As shown in Fig. 6, this important aspect has already been addressed in the manuscript, and we believe that it is evident to the readers, including the reviewer.

Fig. 6 **c** PSC distribution depending on the number of pulses and retention time, obtained from 8×8 pixels of CH-P-integrated memristor array, for color discrimination. Circles and error bars represent average and standard deviation. **d** PSC distribution depending on the number of pulses, obtained from CH-P-integrated memristor under RGB-light pulses with various light intensities from 1 to 20 mW cm^{-2} (RGB pulse conditions: 0.5 s width, and 2 s interval). **f** Dynamic range of CH-P-integrated memristor for color-distinction under various light intensities from 1 to 20 mW cm^{-2} .

Condition 2. The PSC value difference between each state of the B color should be much larger than the highest PSC value of the states of the G and R colors. Similarly, the PSC value difference between each state of the G color should be larger than the highest PSC value of the states of the R color.

In the following Table 1, we present an illustrative example of a 4-bit resolution RGB color-distinguishing case, based on the performance characteristics of the CH-P memristor, including its spectral responsivity and relaxation time. Firstly, the PSC range of each RGB color is adequately separated to avoid any overlap, as suggested in condition 1. Secondly, the PSC difference of each state of the blue color (7.5 mA cm^{-2}) is considerably larger than the highest PSC value of the states of the green and red colors (4.2 and 0.15 mA cm^{-2} , respectively), satisfying condition 2. Similarly, the PSC difference of each state of the green color (0.2 mA cm^{-2}) is notably larger than the highest PSC value of the state of the red color (0.15 mA cm^{-2}).

Supplementary Table 5 4-bit resolution RGB color-distinguish case of CH-P-integrated memristor.

Blue			Green			Red		
State	PSC after 50 th pulse (mA cm^{-2})	Standard deviation (mA cm^{-2})	State	PSC after 50 th pulse (mA cm^{-2})	Standard deviation (mA cm^{-2})	State	PSC after 50 th pulse (mA cm^{-2})	Standard deviation (mA cm^{-2})
15	210.0	$\pm 1.2 \times 10^{-2}$	15	4.2	$\pm 3.9 \times 10^{-4}$	15	0.15	$\pm 8.9 \times 10^{-6}$
14	202.5		14	4		14	0.14	
13	195.0		13	3.8		13	0.13	
12	187.5		12	3.6		12	0.12	
11	180.0		11	3.4		11	0.11	
10	172.5		10	3.2		10	0.10	
9	165.0		9	3		9	0.09	
8	157.5		8	2.8		8	0.08	
7	150.0		7	2.6		7	0.07	
6	142.5		6	2.4		6	0.06	
5	135.0		5	2.2		5	0.05	
4	127.5		4	2.0		4	0.04	
3	120.0		3	1.8		3	0.03	
2	112.5		2	1.6		2	0.02	
1	105.0		1	1.4		1	0.01	
0	0	0	0	0	0			

Under the given condition, our current synapse device can distinguish a 4-bit resolution RGB color image within the range of $(R, G, B) = (0\sim 15, 0\sim 15, 0\sim 15)$. For instance, when we employ a light pulse with $(R, G, B) = (7, 11, 3)$, the output PSC of the CH-P memristor, after 50 pulses, is measured as $123.47 \text{ mA cm}^{-2}$, represented by the sum $(0.07 + 3.4 + 120.0) \text{ mA cm}^{-2}$. In other words, by analyzing the PSC output of $123.47 \text{ mA cm}^{-2}$, we can discern the individual color components as $(R, G, B) = (7, 11, 3)$. Similarly, in another example, when the output PSC of the CH-P device, after 50 pulses, is determined to be $167.32 \text{ mA cm}^{-2}$, we can separate the corresponding (R, G, B) values to $(12, 5, 9)$ as $(0.12 + 2.2 + 165.0) \text{ mA cm}^{-2}$.

Condition 3. To ensure the aforementioned color-separation is valid, it is necessary that the deviation of the PSC value for the B color at each state should be smaller than the interval between states of the G and R colors, as depicted in the following figure. Additionally, the deviation of the PSC value for the G color at each state should also be smaller than the interval between states of the R color. The standard deviation values of BGR in Supplementary Table 5, as provided above, satisfy these requirements.

The deviation of the PSC value at each state for the B color, approximately 0.01 mA cm^{-2} , is comparable to the lowest PSC value of the states of the R color. Therefore, there is a possibility that distinguishing the weakest R signal could be challenging. However, it is important to note that this work serves as a proof-of-concept for the photonic synapse, showcasing its color-distinguishing characteristics even with a simple two-terminal memristor architecture. Moreover, as emphasized previously, this approach is not limited to a specific medium, making it widely applicable. We believe that by designing molecules with enhanced excited state dipole moment, improved spectral responsivity and relaxation time, we can further improve the resolution by optimizing the conductance gap between each state of each color. Consequently, this approach could serve as a platform technology for color-distinguishing synapses utilizing a simple two-terminal architecture, even with the benefit of universality and versatility.

Our modification to the manuscript: Please refer to main text in page 20-21, Fig. 7a,b,c, Supplementary Fig. 37, Supplementary Note 10, and Supplementary Table 5.

Comment #2: *The author suggests that negative weight values can be achieved by utilizing a pair of memristors with comparators. By incorporating the proposed optical programming, each pixel containing two photonic memristors can represent both positive and negative weight values. Since a general weight matrix exhibits high intensity contrast between neighboring pixels (resulting in a noisy image when plotted), each programming beam needs to access every pixel with various intensities. The author proposed waveguide and plasmonic approaches as alternatives to achieve this, which would require microfabrication to pattern the waveguide and non-contact terminals in proximity, similar to the process of patterning metal lines for electrical programming.*

Regarding the "high spatiotemporal resolution" capability of optical programming, it is important to note that according to Reference 20 in the manuscript, this term is applicable primarily to biological neurons, such as in optogenetics. In the case of solid-state semiconductor circuitry, however, electrical programming still provides a reliable means of quantization to represent multiple bits, particularly for memristors. Recent memristor crossbars have achieved 2048 states by employing electrical programming protocols.[1]

[1] Rao, Mingyi, et al. "Thousands of conductance levels in memristors integrated on CMOS." *Nature* 615.7954 (2023): 823-829.

Our response: Thank you for your valuable comment. We agree on your opinion that electrical programming provides a reliable means of quantization to represent multiple bits, particularly for memristors. We believe that both optical and electrical programming approaches have their own merits. Based on your input, we newly cited the reference that you recommended and have made the necessary modifications in the manuscript to highlight the advantages of electrical programming, too.

Our modification to the manuscript: Please refer to main text in page 21.

"The conventional electrical programming offers a reliable means of quantization to represent multiple bits for memristors. For instance, recently, memristor crossbars have achieved up to 2048 conductance states using electrical programming protocols [Rao, M. et al. Thousands of conductance levels in memristors integrated on CMOS. *Nature* 615, 823 (2023)]. However, the optical neural network system provides additional advantages, allowing for higher spatiotemporal resolution and simplification of the weight-update methodology in the hardware array. Moreover, optical programming holds promise for ultrafast speeds, low energy consumption, and large bandwidths by utilizing photons. The utilization of light-irradiation as an additional non-contacting terminal for data-writing can reduce power loss from electrical interconnects during the electrical programming."

Comment #3: *Considering the uncertainty of the color-distinguish capability of the proposed device, it is not possible to clearly pre-process a full 8-bit RGB image at the current stage of development. Therefore, it is recommended to compare the classification results using different approaches. One approach would involve partial preprocessing based on the requiring simulated device limitations, allowing for a certain percentage of overlap. The other approach would involve worse percentage overlaps purely in software.*

Our response: Thank you for your feedback. As we mentioned in response to your comment #1, our current synapse device can distinguish 4-bit resolution RGB color images within the range of (R, G, B) = (0~15, 0~15, 0~15), based on the performance characteristics of the CH-P memristor, although this approach has the potential to be extended to a system that can separate full 8-bit resolution RGB images for the pre-processing with further study. To address this, we made the following modifications to the manuscript:

Firstly, we adjusted the pre-processing results of the CH-P memristor array for CIFAR-10 images to be based on 4-bit resolution RGB color-separation characteristics, as demonstrated in Supplementary Fig. 37.

Secondly, the image recognition tasks for the CIFAR-10 dataset with the convolutional neural network (CNN) were re-performed using the separated RGB three-channel images with 4-bit resolution multiple states. As depicted in Fig. 7b,c, even though the resolution of the image decreased to 4-bit, the accuracy of the image recognition task remained remarkably high, reaching 96% (unidirectional) and 94% (bidirectional).

Additionally, to explore cases with a certain percentage of overlap, as you recommended, we provided exemplary RGB channels where the overlapped RGB components account for 10%, 30%, and 50% (Supplementary Fig. 37). The accuracy results of the CIFAR-10 color image recognition tasks using CNN with the overlapped RGB images are also included in Fig. 7b,c and Supplementary Fig. 38

The image recognition task rates of the CIFAR-10 dataset after pre-processing with the memristors having a full 8-bit resolution RGB color-separating capability are represented in Supplementary Fig. 38 for reference.

Our modification to the manuscript: Please refer to main text in page 21-22, Fig. 7b,c, Supplementary Fig. 37, and Supplementary Fig. 38.

Comment #4: The author has dedicated significant efforts to describing the experimental methodology. One original question was the stability of the fabricated organic devices. It is crucial to understand whether these devices exhibit stability under ambient condition. Additionally, information regarding the lifetimes of the organic devices in the ambient condition would be valuable to ascertain.

Our response: Thank you for your comment. As stated in the manuscript, all measurements were conducted under ambient air conditions. Furthermore, we already extensively characterized the retention behavior at both high-resistance state (HRS) and low-resistance state (LRS), as you suggested before, and we confirmed that our device could maintain data for over 200 hours in ambient conditions.

In response to your current suggestion, we further evaluated the long-term stability of the device under ambient conditions. To achieve this, we conducted additional tests to characterize the retention property of the CH-P-integrated memristor device at elevated temperatures (Supplementary Fig. 30a). Considering the memristor's ion dynamics, activation energy, and reactivity, we evaluated the long-term retention stability at room temperature by fitting the retention time results at higher temperature conditions to the Arrhenius plot (*Sci. Adv.* 2022, 8, eabj7866; *Adv. Funct. Mater.* 2023, 33, 2213064). After the retention tests at higher temperatures, we further examined the D.C I - V sweep cycles (Supplementary Fig. 30b) and light responsivity of the device (Supplementary Fig. 30c) to ensure that it maintained similar performance and trends as before the retention test. As shown in the plots below, the device demonstrated excellent stability, indicating that it can withstand more than 10 years at room temperature without any significant degradation in its performance.

Supplementary Fig. 30 a Plots for retention of high temperature fitted to Arrhenius plot. b D.C I - V sweep cycles before and after 120 °C retention test. c Light responsivity after 120 °C retention test.

Our modification to the manuscript: Please refer to main text in page 13 and Supplementary Fig. 30.

Comment #5: *The author has made the revised manuscript more readable by addressing run-on and lengthy sentences. However, there are still a few instances where sentences contain an excessive number of phrases connected by multiple commas and "and." Therefore, it is recommended to split the information in these complex sentences for better clarity.*

Our response: Thank you for your comment. We further corrected manuscript to improve the readability, as you suggested.

REVIEWERS' COMMENTS

Reviewer #2 (Remarks to the Author):

The authors have largely addressed my concerns. I suggest accepting this manuscript as is.